# Mathematical modeling to understand the role of bivalent thrombin-fibrin binding during polymerization

**Michael A. Kelley, Karin Leiderman** [¤a¤b]*

Department of Applied Mathematics and Statistics, Colorado School of Mines, Golden, Colorado, United States of America

¤a Current address: Department of Mathematics, University of North Carolina at Chapel Hill, Chapel Hill, North Carolina, United States of America
¤b Current address: Computational Medicine Program, University of North Carolina at Chapel Hill, Chapel Hill, North Carolina, United States of America

* karinlg@unc.edu

**Data Availability Statement:** Code required to reproduce the results presented in the manuscript are available at https://github.com/makinnc/KelleyLeidermanPolymerizationCode.

## Abstract

Thrombin is an enzyme produced during blood coagulation that is crucial to the formation of a stable clot. Thrombin cleaves soluble fibrinogen into fibrin, which polymerizes and forms an insoluble, stabilizing gel around the growing clot. A small fraction of circulating fibrinogen is the variant $\gamma_A/\gamma'$, which has been associated with high-affinity thrombin binding and implicated as a risk factor for myocardial infarctions, deep vein thrombosis, and coronary artery disease. Thrombin is also known to be strongly sequestered by polymerized fibrin for extended periods of time in a way that is partially regulated by $\gamma_A/\gamma'$. However, the role of $\gamma_A/\gamma'$-thrombin interactions during fibrin polymerization is not fully understood. Here, we present a mathematical model of fibrin polymerization that considered the interactions between thrombin, fibrinogen, and fibrin, including those with $\gamma_A/\gamma'$. In our model, bivalent thrombin-fibrin binding greatly increased thrombin residency times and allowed for thrombin-trapping during fibrin polymerization. Results from the model showed that early in fibrin polymerization, $\gamma'$ binding to thrombin served to localize the thrombin to the fibrin(ogen), which effectively enhanced the enzymatic conversion of fibrinogen to fibrin. When all the fibrin was fully generated, however, the fibrin-thrombin binding persisted but the effect of fibrin on thrombin switched quickly to serve as a sink, essentially removing all free thrombin from the system. This dual role for $\gamma'$-thrombin binding during polymerization led to a paradoxical decrease in trapped thrombin as the amount of $\gamma'$ was increased. The model highlighted biochemical and biophysical roles for fibrin-thrombin interactions during polymerization and agreed well with experimental observations.

## Author summary

We developed a mathematical model of fibrin polymerization that explicitly incorporated thrombin-fibrin(ogen) interactions, including the $\gamma'$ variant of fibrin(ogen). This variant

**Funding:** This work was supported by the the Army Research Office (ARO-12369656 to MK and KL), the National Science Foundation (CAREER DMS-1848221 to MK and KL), and the National Institutes of Health (R01 HL151984, R01 HL120728 to KL). The funders had no role in study design, data collection and analysis, decision to publish, or preparation of the manuscript.

**Competing interests:** The authors have declared that no competing interests exist.

is associated with clots that are more resistant to fibrinolysis and is therefore implicated as a risk factor in cardiovascular disease. However, the underlying mechanism it is not completely understood. We previously modeled $\gamma'$-thrombin interactions in a preformed fibrin clot and hypothesized that some thrombin became physically trapped within the clot during its formation (fibrin polymerization). We developed the model in this study to test that hypothesis and determine the role of $\gamma'$ during polymerization. The new model showed the plausibility of large amounts of thrombin being trapped within fibrin fibers during polymerization and suggested a dual role for $\gamma'$ during polymerization: localization of thrombin to the fibrin(ogen) during an initial early phase of polymerization and sequestration of thrombin during the later phase. The model results suggested a new regulatory mechanism for fibrin polymerization.

## Introduction

When a blood vessel is injured, the human body invokes the hemostatic response and forms a blood clot to prevent bleeding. The response includes both platelet aggregation and blood coagulation. At the site of an injury, the exposed subendothelium of the blood vessel puts collagen and tissue factor, initiators of the clotting process, in contact with the flowing blood. Platelets bind to collagen at the injury site, become activated, and aggregate to form a platelet plug. Simultaneously, tissue factor stimulates the complex biochemical system of reactions that comprise blood coagulation. Coagulation is mediated by multiple positive and negative biochemical feedback loops that take place at the subendothelium, in the plasma, and on platelet and other blood-cell surfaces. Coagulation results in the production of thrombin, a major enzyme in the clotting process and common indicator of overall clotting potential. Thrombin acts on various substrates in coagulation, both amplifying and inhibiting its own production. One important substrate for thrombin is the soluble protein fibrinogen; thrombin enzymatically cleaves fibrinogen into the insoluble form, fibrin, which polymerizes into a gel mesh that provides structure and stability to the growing blood clot.

Thrombin is a serine protease with an active site and two additional binding sites that help anchor thrombin to platelets and fibrin(ogen) during enzymatic activity [1–5]. These binding sites are called exosite 1 and exosite 2 and they are positioned on either side of the active site. Fibrin(ogen) is symmetric molecule with a central E-domain flanked by two D domains. Each E-domain has two binding sites for thrombin exosite 1 and each D domain has a $\gamma$ chain. About 15% of the circulating fibrinogen in plasma is in a minority conformation, called $\gamma'$ fibrinogen, [6]. The $\gamma$-chains of $\gamma'$ fibrin(ogen) possess an additional binding site to which exosite 2 of thrombin can bind [6]. Thrombin molecules can potentially bind bivalently to both an E-domain and a $\gamma'$ binding site, via both exosites, bridging two fibrin monomers if they are close enough to one another [7]. Note that this bivalent binding likely cannot occur on one single fibrin monomer as the E-domain and $\gamma'$-chain are too far apart [8].

Thrombin is required for fibrin polymerization. When thrombin binds via exosite 1 to the central E-domain of fibrinogen, it cleaves fibrinopeptide A (FpA), and the remaining molecule is often called fibrin I [9]. In a step sequential to FpA cleavage, E-domain-bound thrombin cleaves fibrinopeptide B (FpB), forming fibrin II [9]. Fibrin I and II bind to each other to form oligomers, where fibrin I and II are bound end-to-end in a half-staggered formation [10]. These oligomers grow in length by binding additional fibrin I and II monomers or by combining with other oligomers to form even larger oligomers [11]. Once an oligomer has reached some critical length (sometimes assumed to be about 11 monomers [11, 12]) it becomes a

protofibril. Protofibrils can grow in length by binding fibrin monomers, oligomers, and other protofibrils. They can also aggregate laterally and form new fibers, or attach laterally to an existing fiber, increasing the width of that fiber [11, 13].

During and after polymerization and clot formation, thrombin interacts with fibrinogen and fibrin. The latter is thought to be an inhibitory mechanism that prevents thrombin from participating in other reactions, hence the original name for fibrin being antithrombin I [14]. The dynamic thrombin binding interactions during polymerization, and how they may affect polymerization more broadly, are not completely understood.

Thrombin associates with $\gamma'$ fibrin(ogen) bivalently [7] but also allosterically [15] and both of these types of interactions could modify thrombin's enzymatic activity and binding behavior in the presence of $\gamma'$ fibrinogen. This could play an important role in cardiovascular health as $\gamma'$ fibrinogen has been implicated as a risk factor for myocardial infarctions [16], deep vein thrombosis [17], and coronary artery disease [18]. The interactions between thrombin and $\gamma'$ fibrinogen may also provide possible avenues to develop targeted, thrombin-exosite peptides that exploit thrombin allostery [15, 19, 20]. Low levels of $\gamma'$ fibrin(ogen) are associated with increased fibrin generation and increased thrombin generation [21]. Additionally, it has been suggested that fibrin-trapped thrombin, or the disruption thereof, could play a role in the observed fibrinolytic breakdown as a result of COVID-19 infection that leads to unintended bleeding and clotting. [22].

Thrombin was first suggested to become physically trapped inside fibrin clots by the experimental work of Banninger et al., [23], and later potentially inside the fibrin fibers in our previous mathematical modeling work [24]. Fibrin sequesters thrombin for extended periods of time, protecting it from physical removal and chemical inhibition [7, 25, 26]. This phenomenon is observed across a wide range of conditions including scenarios where thrombin is incorporated during fibrin clot formation [7], incorporated after fibrin clot formation [25], and in both purified [7, 25] and whole blood systems [26]. Zhu et al. suggested that thrombin sequestration may be a result of a molar excess of fibrin binding sites [26] and our previous work suggested that thrombin was potentially physically trapped [24]. Likely, it is due to be some combination of these effects, each playing a different role under varying circumstances, but this is also not completely understood.

Sequestered thrombin could have an inhibitory role by limiting the spread of thrombin and subsequent clotting [27]. It could also stabilize a blood clot, providing a new source of thrombin upon clot damage, removing the need for re-initialization of the entire clotting process [28]. It could be pathological in nature, with released thrombin causing off-target clotting away from the injury site [29, 30]. Additionally, it may slow fibrinolysis via thrombin activatable fibrinolysis inhibitor (TAFI) activation as it is released during clot dissolution. Although we could not determine the precise role for thrombin sequestration at the physiological level, a major goal of this study was to address questions regarding the role of $\gamma'$ fibrin(ogen) and thrombin's bivalent binding during the polymerization process, since this is what leads to the subsequent thrombin sequestration. One might guess that if $\gamma'$ acts purely as a high-affinity sink, then increasing $\gamma'$ fibrin(ogen) in polymerization experiments would result in fibrin that is characteristically similar to that formed when decreasing thrombin concentrations. However, the exact opposite observations have been made [31, 32]! Part of this study was to use a mathematical modeling approach to try to understand and potentially elucidate a mechanism of this paradoxical behavior.

Mathematical models of fibrin polymerization have previously been developed that we built on, extended, and/or integrated. Weisel and Nagaswami [11] developed the first comprehensive continuum model of polymerization that simulated the activation of fibrinogen monomers into fibrin monomers, assumed to occur at a constant rate, and subsequent formation of

oligomers, protofibrils, and fibers. Resulting dynamic fiber characterizations (quantified by number of protofibrils per fiber, called protofibril number) from this model qualitatively agreed with experimental turbidity data. One of the many scenarios modeled in that study showed that as thrombin increased, the width of fibrin fibers decreased. The model results reinforced the idea that clot structure was kinetically controlled. This innovative model did have a few limitations; it did not explicitly consider thrombin but rather used a constant rate parameter that varied with assumed levels of thrombin and thus the model could not track dynamic thrombin-fibrinogen interactions during fibrin I formation nor could it discriminate between fibrin I and fibrin II.

The first kinetic models of the conversion of fibrinogen to fibrin were developed by Lewis, Naski, and Shafer [9, 33]. The models considered formation of both fibrin I and fibrin II, and helped support the data and the idea that there is an ordered, sequential release of FpA and FpB. Fibrinogen conversion by thrombin was assumed to obey Michaelis-Menten kinetics and thrombin-fibrinogen interactions were not explicitly modeled. One model explicitly accounted for thrombin and included antithrombin inhibition of thrombin [9]. It is now accepted that FpA cleavage allows for oligomerization, and that FpB cleavage may enhance lateral aggregation of protofibrils [34–36]. However, the exact mechanism(s) through which thrombin, $\gamma'$, fibrin I, and fibrin II interact while forming oligomers, protofibrils, and fibers remain unclear.

Allan and colleagues showed that negatively charged $\gamma'$-chains cause steric interference that slows lateral aggregation of protofibrils [37]. They suggested this mechanism over thrombin-$\gamma'$ binding to cause longer lag times and lower overall turbidity in experiments comparing $\gamma_A/\gamma'$ to $\gamma_A/\gamma_A$ fibrinogen. This conclusion was based on experiments with both thrombin and reptilase, an enzyme that binds to fibrin in a similar way to thrombin, but does not cleave FpB, thus leaves only fibrin I to polymerize. This was taken to mean that thrombin-$\gamma'$ binding was not responsible for the differences in fibrin structure between the $\gamma_A/\gamma_A$ and $\gamma_A/\gamma'$ cases. Gersh et al. showed that increasing the ratio of $\gamma'$ chains to E-domains decreased the number of protofibrils per fiber while simultaneously speeding up FpB release [31]. Since FpB is cleaved by thrombin, this suggests that thrombin may be directly influencing the fibrin structure through the $\gamma'$ binding and possibly the bivalent mediated FpB cleavage. Allan and colleagues suggest that the effects of $\gamma'$ on fiber structure are solely due to steric interference and are independent of thrombin-$\gamma'$ interactions. Other work has shown that $\gamma'$ significantly affects the rate of FpB release and the formation of fibrin II, nicely summarized in Gersh et al. [31]. Further, since reptilase prevents the formation of fibrin II, the studies by Allan et al. do not rule out the possibility of structural differences due to $\gamma'$ mediated FpB release.

The experiments described so far were turbidity measurements at fixed wavelengths, with varied fibrin(ogen) type and thrombin concentrations, and performed on timescales ranging from minutes to a few hours [9, 11, 31, 37]. Domingues and colleagues monitored fibrin polymerization over 24 hours, also using different fibrinogen types and varied thrombin concentrations, but measuring the final turbidity over a range of different wavelengths [32]. With this methodology, they were able to report quantitative structural measurements, including fiber radius, protofibril number, protein density, and distance between protofibrils.

To describe the results from Domingues et al., and our results that follow later in this paper, it is instructive to describe the various types and combinations of fibrin(ogen) type that were used in the experiments. Fibrin(ogen) is defined by the ratio of the $\gamma'$ chains to E-domains for each monomer, i.e., 0:1 means there were no $\gamma'$-chains, 1:1 means there was one $\gamma'$ and one $\gamma_A$ per monomer, and 2:1 means that both chains on a monomer were $\gamma'$ [7, 25, 32, 37]. Fractionated $\gamma_A/\gamma_A$ fibrin(ogen) has only E-domain (0:1), wild-type or unfractionated $\gamma_A/\gamma'$ fibrin (ogen) has approximately 0.3 times as many $\gamma'$ chains as E-domains assuming that about 15% of fibrinogin is the $\gamma'$ variant (0.3:1), fractionated or recombinant $\gamma_A/\gamma'$ fibrin is such that every

fibrin has a single $\gamma'$ chain (1:1), and recombinant $\gamma'/\gamma'$ fibrin is where every $\gamma$ chain is a $\gamma'$ chain (2:1).

Domingues and colleagues found that while increasing thrombin added to $\gamma_A/\gamma_A$ fibrinogen, the radius of the resulting fibrin fiber decreased, but only by about 15% over a 4 order of magnitude change in thrombin concentration. For the $\gamma_A/\gamma'$ case they observed no appreciable change in fiber radius over a wide range of thrombin concentrations. They reported the protofibril number per fiber for $\gamma_A/\gamma_A$ to be much higher than that of $\gamma_A/\gamma'$ at thrombin concentrations strictly less than 1 U/ml (10 nM) and found similar behavior in protein densities and the distances between protofibrils. All together, these results suggested a relationship between thrombin and $\gamma'$ that had a significant effect on fibrin fiber structure at low thrombin concentrations. For thrombin concentrations 1 U/ml and above, the differences between these structural metrics for $\gamma_A/\gamma_A$ and $\gamma_A/\gamma'$ were almost negligible. This last result is somewhat at odds with their own previous studies (Allan et al., 2012, described above) where large differences in turbidity curves were observed between $\gamma_A/\gamma_A$ and $\gamma_A/\gamma'$ fibrinogen initiated with 1 U/ml thrombin [37]. The earlier study did utilize a lower fibrinogen concentration than that used in the Domingues et al. study and additionally used a FXIII inhibitor.

In a study focused on fibrinolysis speeds of fibers comprised of $\gamma_A/\gamma_A$ versus $\gamma_A/\gamma'$ fibrinogen, Kim and colleagues also measurements of clotting times for each of the different fibrinogen types via turbidity, using 1 nM thrombin and fibrinogen concentrations ranging from $2 - 18 \mu M$ [38]. Clot times were defined as the time to the half-maximal turbidity measurement. Across their range of fibrinogen concentrations, $\gamma_A/\gamma'$ fibrin had longer clot times that continually increased as fibrinogen was increased, from about 4 minutes to a max of about 8 minutes. The clot times for $\gamma_A/\gamma_A$ fibrinogen also increased as fibrinogen concentration increased, but in a saturable manner, from 3 to 5 minutes. The ratio of the clot times for the two fibrinogen types was thus non-monotonic, increasing and then decreasing as the fibrinogen concentration increased. These experiments were repeated using reptilase (batroxobin) instead of thrombin, which resulted in shorter clot times for both fibrinogen types and a linear increase in the ratio of clot times for the two fibrinogen types. Overall these data show that increasing the ratio of $\gamma'$ binding sites to and E-domains or, the total number of thrombin binding sites present, results in longer clot times [38].

Banninger and colleagues were the first to show existence of an irremovable population of thrombin trapped inside fibrin clots [23]. Our recent work further proposed that thrombin may become physically trapped inside of individual fibrin fibers [24]. To better understand the dynamics of this trapping during polymerization and determine if the trapping was feasible, i.e., occurring on the appropriate timescales during polymerization, we developed a stochastic model to monitor single thrombin molecules within fibrin junctions. In the current study, we estimated the times that thrombin remained in a bound state in these junctions and showed that they were long enough for physical trapping during polymerization to be feasible. This provided motivation to develop a continuum model to predict the quantity of this trapped thrombin; we extended and integrated previous mathematical models of fibrin polymerization to include dynamic thrombin-dependent fibrinopeptide cleavage and thrombin-fibrin binding. Our new integrated model also allowed for the investigation of $\gamma'$ fibrin(ogen)'s role in polymerization. We found that $\gamma'$ fibrin(ogen) plays a dual role in the polymerization process: early in polymerization, the high affinity, $\gamma'$-mediated thrombin binding served to localize thrombin to protofibrils where it cleaved FpB, and once the FpB had been depleted, the $\gamma'$-mediated bivalent binding served to keep thrombin tightly bound to fibrin. These results suggest a mechanism underlying paradoxical experimental observations and may provide insight into the role(s) of thrombin-fibrin interactions and thrombin sequestration during polymerization and clotting more broadly.

**Table 1. Bivalent binding reactions and kinetic constants.**

| # | Reaction | Dissociation Constant & Literature Values | | Kinetic Rates |
|---|----------|-------------------------------------------|---|---------------|
| 1 | $T + E1 \rightleftharpoons B_{E1}$ | $K_{d,E}$ | $2.8\mu$M | $k_{on,E}, k_{off,E}$ |
| 2 | $T + E \rightleftharpoons B_E$ | $K_{d,E}$ | $2.8\mu$M | $k_{on,E}, k_{off,E}$ |
| 3 | $T + G1 \rightleftharpoons B_{G1}$ | $K_{d,G}$ | $9\mu$M | $k_{on,G}, k_{off,G}$ |
| 4 | $T + G \rightleftharpoons B_G$ | $K_{d,G}$ | $9\mu$M | $k_{on,G}, k_{off,G}$ |
| 5 | $B_E \rightleftharpoons B$ | $\hat{k}_{on,G2}$ | sampled from estimated probability distribution [24] | $k_{on,G2}, k_{off,G}$ |
| 6 | $B_G \rightleftharpoons B$ | $\hat{k}_{on,L2}$ | sampled from estimated probability distribution [24] | $k_{on,E2}, k_{off,E}$ |

## Methods

### Bivalent binding scheme

The bivalent thrombin-fibrin binding scheme used throughout this study is from our previous work [24] and is described in Table 1. Here, $T$ is free thrombin, $B_{E1}$ is thrombin bound to the E-domain, not in proximity to a nearby $\gamma'$ binding site, $B_E$ is thrombin bound to the E-domain and in proximity to a nearby $\gamma'$ binding site, and $B_G$ is thrombin bound to the $\gamma'$ binding site near an E-domain. In the polymerization model only, we also consider $B_{G1}$, which is thrombin bound to a $\gamma'$ binding site that is not in proximity to an E-domain, which occurs before a fibrin monomer has been incorporated into an oligomer. Finally, $B$ is thrombin bound bivalently to both an E-domain binding site and a $\gamma'$ binding site. All kinetic rates used for the bivalent binding are those estimated in our previous work [24].

### Stochastic Binding Model (SBM)

A stochastic simulation algorithm (SSA), or Gillespie algorithm, [39] was used to investigate a single thrombin molecule binding to fibrin binding sites under two distinct conditions. A continuous-time Markov process was constructed and simulated using the SSA. First, the case of thrombin binding to fractionated $\gamma_A/\gamma_A$ fibrin(ogen) was considered (Fig 1A). There was no $\gamma'$ binding so thrombin could bind only to the E-domain (2 binding sites). We refer to this as an 'A-A' binding junction (AA). Second, thrombin was considered to bind to fractionated $\gamma_A/\gamma'$ fibrin (Fig 1B). The half-staggered formation of the fibrin monomers allowed for the bivalent binding of thrombin to fibrin via both exosites to both the E-domain and $\gamma'$-chain. In this case every junction on binding sites had a $\gamma'$ binding site, called an 'A-P' binding junction (AP). These two cases represent two fundamental cases of thrombin-fibrin binding, with and without bivalent binding (no $\gamma'$ sites). A weighted average of the results from both cases were then used to estimate results for the wild-type case, where 15% of the $\gamma$ chains are $\gamma'$ chains, a mix of AA's and AP's.

Consider the general state variable $X = (X_1, X_2, X_3, .., X_n)$, where $X$ represents the possible states of a thrombin molecule while binding to an AA or AP, and $X_i(t)$ represents the number of molecules in state i, at time t [39]. A simplified case was considered where we tracked only a single thrombin molecule at a time. Thus, at each point in time, exactly one $X_i = 1$ and all other $X_i = 0$ and we needed only to consider and track the non-zero $X_i$. Through time, a single vector of states, $X_k$, and the time to each transition, $t_k$ were tracked, where k represents the $k^{th}$ step in the algorithm. For each state, $X_k$, the transition rates, $\lambda_i(X_k)$, are the rates at which the thrombin molecule moved from the state it was in, $X_k$, to a new state $X_{k+1}$. A simplified version of the "direct method" for simulating the Gillespie algorithm was used [39]:

1. Start in the initial state $X_0 = X_k$ for some initial time $t = t_0 = t_k$

2. Draw random numbers, $r_1$ and $r_2$, from a standard uniform distribution, $U(0,1)$.

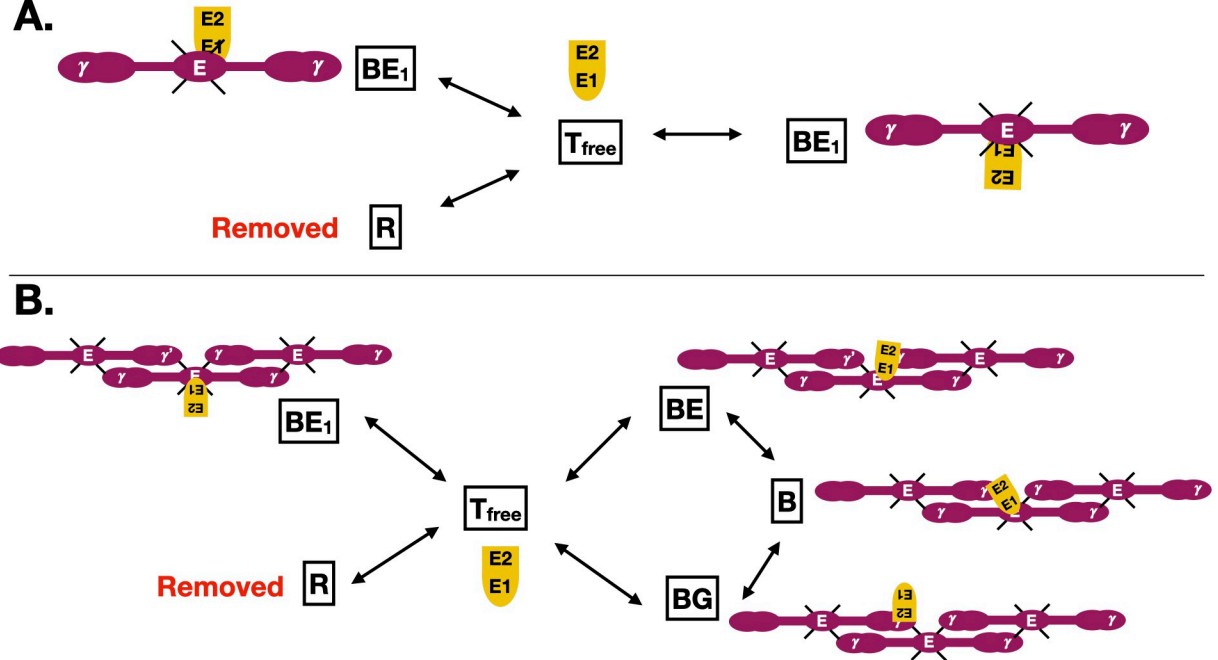

**Fig 1. Schematic of single molecule stochastic binding model.** (A) A single thrombin molecule binding at a junction of fibrin monomers without bivalent binding (AA junction). Thrombin can be unbound ($T_{free}$), bound to fibrin via two E-domain binding sites ($BE_1$), or removed from the system (R), which is an absorbing (end state). Thrombin is in yellow, fibrin in purple, and red is the removed state. (B) A single thrombin molecule binding at a junction of fibrin monomers with bivalent binding (AP junction). Thrombin can be unbound ($T_{free}$), bound to fibrin via two E-domain binding sites ($BE_1$ and BE), bound to a single $\gamma'$ binding site (BG), or can transition from BE and BG to the bivalently bound state (B), or can be removed from the system to state (R), an absorbing (end state).

3. Calculate the transition rates $\lambda_i(X_k)$ and the sum, $\lambda = \sum_{i=1}^{n} \lambda_i(X_k)$

4. Simulate time:

   - Draw a time, $t_k$, from an exponential distribution with a mean of $1/\lambda$

   - Calculate $t_{k+1} = \frac{-log(r_1)}{\lambda}$

5. Choose which state thrombin will transition to by performing a CDF inversion [40]

   - Compare $r_1$ with $\sum_{i=1}^{j} \lambda_j(X_k)/\lambda$

   - if $r_2 < \lambda_1(X_k)/\lambda$ then $X_{k+1} = X_1$

   - And in general: if $\sum_{i=1}^{j} \lambda_i(X_k)/\lambda < r_2 < \sum_{i=1}^{j+1} \lambda_i(X_k)/\lambda$ then $X_{k+1} = X_{j+1}$

6. update time: $t = t + t_k + 1$

7. Repeat steps 2–6 until a stopping condition is met

   For the AA case, consider the states $(X_1, X_2, X_3) = (R, T, B_{E1})$ (Fig 1A), where thrombin can be free in state $T_{free}$, bound to the E-domain, state $B_{E1}$, or removed from the system, R. The removed state is an absorbing state and is the stopping criterion, meaning that when the thrombin molecule transitions to R, that particular simulation is finished. The transition rates for the model were derived from Table 1. As an example, the transition rates from state T are as follows: $\lambda_{BE1}(T) = k_{on,E} \cdot F_E$, and $\lambda_R(T) = k_r$, where $k_r$ is some removal rate (1/s), and $F_E$ and

$F_G$ are the total number of E-domain and $\gamma'$ binding sites, respectively. Each simulation was initiated with one thrombin molecule in the $B_{E1}$ state. The two E-domain sites were accounted for by doubling the transition rate to that bound state.

For the AP case, consider the states $(X_1, X_2, X_3, X_4, X_5, X_6) = (R, T, B_{E1}, B_E, B_G, B)$ (Fig 1B), where thrombin can be free $T_{free}$, bound to the E-domain, $B_{E1}$ and $B_E$, bound to $\gamma'$, $B_G$, transition from $B_E$ and $B_G$ to the bivalently bound state, $B$, or can be removed from the system, $R$, as in the AA case. The transition rates for the model were again derived from Table 1. The rates for the sequential binding step ($B_{E'}$ to $B$ and $B_G$ to $B$) were sampled from our previously estimated joint probability distribution [24]; we used 2650 different sampled parameter pairs for the sequential rates. As an example, the transition rates from state T are as follows: $\lambda_{BE1}(T) = k_{on,E} \cdot F_E$, $\lambda_{BE}(T) = k_{on,E} \cdot F_G$, $\lambda_{BG}(T) = k_{on,G} \cdot F_G$, and $\lambda_R(T) = k_r$, where $k_r$ is some removal rate $(1/s)$, and $F_E$ and $F_G$ are as before.

For both the AA and AP cases, the algorithm was run 2000 times until the thrombin reached the absorbing state. This was repeated for nine values of $k_r$ to represent thrombin binding over a wide range of potential removal rates. We calculated the time spent in any of low affinity bound states, $B_{E1}$, $B_G$, and $B_E$, in the high affinity bound state, $B$, and the time spent susceptible to removal in the free state, $T$.

### Fibrin polymerization and thrombin binding

**Fibrin polymerization model.** We extended and integrated previous models of thrombin-fibrin interactions and fibrin polymerization: the Naski-Schafer model of fibrinopeptide cleavage [9], the Weisel-Nagaswami model of fibrin oligomerization and polymerization [11], and our own previous model of thrombin-fibrin bivalent binding [24]. Our model of thrombin-fibrin binding enables the coupling of the previous two models, allowing for feedback and additional interactions resulting from thrombin-fibrin binding and thrombin's enzymatic cleavage of fibrinopeptides. First we will describe the model at large, describing the species and flow of the model. Then, we will briefly describe these models and our extensions, with a schematic shown in Fig 2. The Naski-Schafer model accounts for fibrin I and I formation only tracking them until they are generically polymerized and removed from the system. We extended this to track the fibrin I and II quantity and the dynamic conversion of fibrin I and II throughout the the entire polymerization process. The Weisel-Nagaswami model was described as a constant activation of fibrinogen into a single fibrin species which formed oligomers. Those oligomers grew linearly, one monomer at a time. This model was the first and only of its kind. We extended the model to include all possible oligomer binding combinations when binding and forming a protofibril or when binding to an existing protofibril and increasing its size. This meant that an oligomer or protofibril could grow in length by any number of monomers up to 10, the critical size at which an oligomer becomes a protofibril.

**Model design.** The model takes as inputs $T$ and $f_{ab}$, thrombin and fibrinogen respectively. Thrombin converts fibrinogen into fibrin I, $f_b$, by cleaving FpA. Thrombin then converts fibrin I into fibrin II, $f$, by cleaving FpB. Fibrin I and fibrin II can both begin to bind with each other form oligomers. Oligomers are chains of fibrin I and II monomers less than 11 monomers in size. These oligomers bind to fibrin I and II and to other oligomers growing in length. Oligomers are denoted as $f_1, f_2, \cdots, f_9$, and $f_{10}$ with the subscript denoting length in monomers. Thrombin continues to bind to fibrin in oligomer form and cleave FpB. Because of this we needed to track the number of fibrin I and fibrin II monomers in various oligomers. We introduced the variables $C_b$ and $C_f$ to track oligomerized fibrin I and fibrin II, respectively. This means that during the model description when we want to denote thrombin moving into or out of one these states as

$$C_b \pm = \text{change in monomers},$$

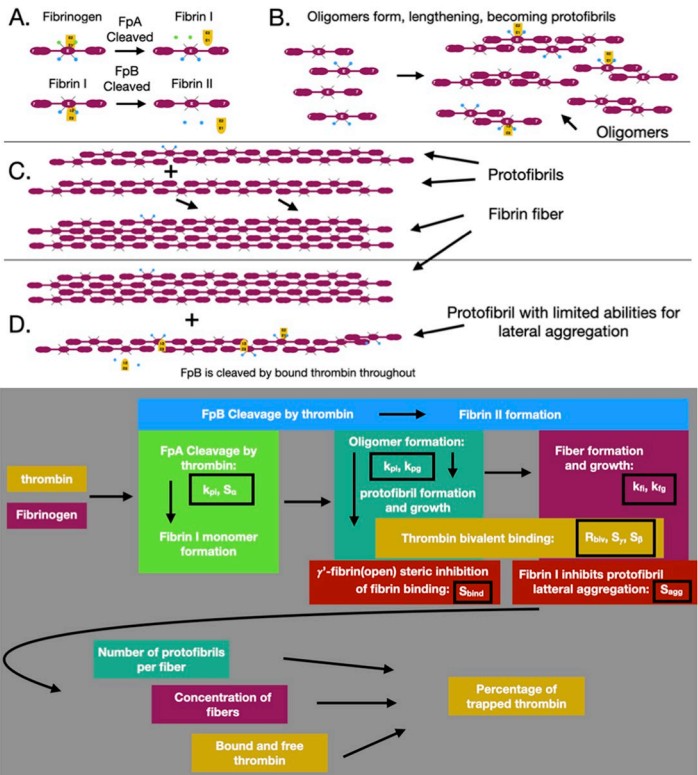

**Fig 2. Schematic of the four main stages in the polymerization model.** (A) Thrombin forms fibrin I and fibrin II monomers. Thrombin (yellow shape) binds the the E-domain of fibrinogen (purple with blue and green dots), cleaving FpA (green dots), which converts fibrinogen into fibrin I (purple with blue dots). Thrombin binds to the E-domain to cleave FpB (blue dots), converting fibrin I into fibrin II (purple with no dots). (B) Fibrin I and II bind together forming half-staggered chains called oligomers. Fibrin I and II also bind to existing oligomers, increasing them in length. When an oligomer reaches a critical length (11 monomers in length for this study), it becomes a protofibril. (C) Protofibrils aggregate laterally, forming fibers, which are cable-like bundles of protofibrils. Additional protofibrils can bind to an existing fiber, increasing its diameter. (D) As protofibrils and fibers are formed, FpB continues to be cleaved from fibrin I that is incorporated into protofibrils and fibers. Fibrin I is considered to have a limiting effect of lateral aggregation, slowing the process as the ratio of fibrin I to fibrin II increases. The bottom of the figure shows a diagram of the model flow, particularly how key components of the model are connected and which estimated parameters affect each component. The colors in the flow diagram match those in the schematics above.

where the sign designates whether the monomers are moving into or out of the $C_b$ category. We use the same notation for monomers moving in or out of $C_f$. This allows us to track and model the continued cleavage of FpB by thrombin. Once an oligomer reached 11 monomers in length, it was considered to be a protofibril, denoted $f_n$. Protofibrils can bind laterally to each other, forming fibers or increasing the thickness of existing fibers. Now we also need to know how much FpB is still present because we account for the slower lateral aggregation of fibrin I as compared to fibrin II. So again, we will use two additional variables to track the number of fibrin I and fibrin II monomers in protofibril form, $C_{fb}$ and $C_{fn}$. We will track the number of fibrin monomers in fibers (one or more protofibril laterally aggregated) as $C_{fr}$. We will use the same ± = notation to describe movement of fibrin monomers in and out of $C_{fb}$, $C_{fn}$, and $C_{fr}$. Thrombin bound to the E-domain of fibrin is denoted as $Ef_{ab}$ or $Ef_b$ if it the fibrin is free and as $BE_1$ or $BE$ if the fibrin is part of an oligomer, protofibril, or fiber.

 **Fibrinopeptide cleavage.** We built upon the work of Lewis et al. [33] and Naski-Shafer [9] to model fibrinopeptide cleavage. We included the same steps for the cleavage and

conversion but instead of Michaelis-Menten kinetics, we used mass action kinetics and a two-step enzyme reaction:

$$f_{ab} + T \xleftrightarrow{K_m^a} Ef_{ab} \xrightarrow{k_{cat,a}} f_b + T, \tag{1}$$

$$f_b + T \xleftrightarrow{K_m^b} Ef_b \xrightarrow{k_{cat,b}} f + T. \tag{2}$$

Here, $f_{ab}$ and $T$ are fibrinogen and thrombin, respectively, and $Ef_{ab}$ is E-domain bound thrombin. The first reaction describes conversion of fibrinogen to fibrin I through the thrombin cleavage of FpA. The second reaction describes the conversion of fibrin I to II through the thrombin cleavage of FpB. All the kinetic constants, the $K_m$ and $k_{cat}$ values, were taken from the original study by Naski and Schafer [9].

**Oligomer & protofibril formation and growth.** We extended and modified the Weisel-Nagaswami model [11] to include the FpA and FpB cleavage with fibrinogen, fibrin I, and fibrin II as described above. The details of how we modeled oligomer and protofibril formation and growth are described below.

**Oligomer formation.** Oligomers can form in several ways, shown schematically in Fig 2B). We assumed that two fibrin monomers bind with rate $k_{pi}$ to form $f_2$. This $f_2$ could be any combination of fibrin I or fibrin II:

$$f_b + f_b \xrightarrow{k_{pi}} f_2 \tag{3}$$

$$f + f \xrightarrow{k_{pi}} f_2 \tag{4}$$

$$f + f_b \xrightarrow{k_{pi}} f_2. \tag{5}$$

To track the amounts of fibrin I and fibrin II in the oligmers, we use the variables $C_b$ and $C_f$, with all fibrin I going to $C_b$ and all fibrin II going to $C_f$. For the initial oligomer formations, fibrin monomers were partitioned as follows:

$$C_b + = 2k_{pi}f_b^2 + k_{pi}f_b \tag{6}$$

$$C_f + = 2k_{pi}f^2 + k_{pi}f. \tag{7}$$

**Oligomer elongation.** Elongation of oligomers included binding of as little as two fibrin monomers with the addition of monomers and other oligomers. Let $f_3, \ldots, f_{10}$ be the half-staggered chains of fibrin monomers bound together, making up oligomers that are 2 to 10 monomers in length. Oligomers could grow linearly, with a single fibrin I or fibrin II binding to and increasing its length by 1, i.e., with fibrin I:

$$f_2 + f_b \xrightarrow{k_{pi}} f_3, \tag{8}$$

$$f_3 + f_b \xrightarrow{k_{pi}} f_4, \\ \vdots \tag{9}$$

$$f_9 + f_b \xrightarrow{k_{pi}} f_{10}. \tag{10}$$

Every time an $f_b$ binds, it is added to $C_b$ to keep count of fibrin I within the oligomers, tracked using $C_b$:

$$C_b+ = 2k_{pi}f_bf_2 + \cdots + 10k_{pi}f_bf_9. \tag{11}$$

Fibrin II also binds to existing oligomers and linearly increased their length:

$$f_2 + f \xrightarrow{k_{pi}} f_3 \tag{12}$$

$$f_3 + f \xrightarrow{k_{pi}} f_4$$
$$\vdots \tag{13}$$

$$f_9 + f \xrightarrow{k_{pi}} f_{10} \tag{14}$$

Similarly, we count each $f$ that binds by adding it to $C_f$:

$$C_f+ = 2k_{pi}ff_2 + \cdots + 10k_{pi}ff_9. \tag{15}$$

We also consider *all* other possible oligomer binding combinations. Let $f_i$ and $f_j$ be two oligomers of length $i$ and $j$ respectively, for $i + j < 11$. We considered the following reactions:

$$f_i + f_j \xrightarrow{k_{pi}} f_{i+j}. \tag{16}$$

Since monomers were not be added or removed, this step did not affect $C_b$ and $C_f$.

**Protofibril formation.**   Once an oligomer reaches the critical length of 11 monomers in length it became a protofibril and was then able to aggregate laterally into fibrin fibers. We will refer to protofibrils as $f_n$. The first and most obvious way for this to happen is for an $f_{10}$ to linearly grow by binding to either $f_b$ or $f$.

$$f_{10} + f_b \xrightarrow{k_{pi}} f_n, \tag{17}$$

$$f_{10} + f \xrightarrow{k_{pi}} f_n. \tag{18}$$

Care must be taken when counting the total fibrin I and fibrin II. When $f_b$ or $f$ binds, we need to subtract from $C_b$ and $C_f$ appropriately and add to $C_{fb}$ and $C_{fn}$ appropriately. We define $\phi_b = \frac{C_b}{C_f+C_b}$, the fraction of oligomer associated fibrin I. We tracked the fibrin I and II monomers using:

$$C_b- = 10\phi_b k_{pi}f_bf_{10} \tag{19}$$

$$C_f- = 10(1 - \phi_b)k_{pi}ff_{10} \tag{20}$$

$$C_{fb}+ = 10\phi_b k_{pi}f_bf_{10} + k_{pi}f_bf_{10} \tag{21}$$

$$C_{fn}+ = 10(1 - \phi_b)k_{pi}ff_{10} + k_{pi}ff_{10}. \tag{22}$$

We consider the fraction of oligomer-bound fibrin I and use it to partition the fibrin monomers in each $f_{10}$ into $C_{fb}$ and $C_{fn}$ according to the fraction of an arbitrary oligomer made of fibrin I and fibrin II. Additionally, we add the fibrin monomers that are binding to the oligomers forming protofibrils, adding them to $C_{fb}$ and $C_{fn}$ respectively.

Protofibrils can also form when oligomers $f_i$ and $f_j$ of length $i$ and $j$, such that $i + j \geq 11$, bind together forming a new protofibril:

$$f_i + f_j \xrightarrow{k_{pi}^N} f_n. \tag{23}$$

We again tracked the fibrin monomers as they moved from oligomers to protofibrils:

$$C_b- = \phi_b \cdot (i+j)(k_{pi}^N f_i f_j) \tag{24}$$

$$C_f- = (1 - \phi_b) \cdot (i+j)(k_{pi}^N f_i f_j) \tag{25}$$

$$C_{fb}+ = \phi_b \cdot (i+j)(k_{pi}^N f_i f_j) \tag{26}$$

$$C_{fn}+ = (1 - \phi_b) \cdot (i+j)(k_{pi}^N f_i f_j). \tag{27}$$

Throughout this process we assumed that the effective $k_{pi}$ decreased as the size of the oligomers binding increased, reflecting that diffusion slows as the size increased, limiting the mass action encounters. Let $k_{pi}^N = k_{pi} \cdot S_{olig}(N)$, then $k_{pi}$ is the base value and $k_{pi}^N$ is the effective binding rate of the two monomers for $i + j = N$. We assumed the following linear scaling function:

$$S_{olig}(N) = \frac{21 - N}{19}. \tag{28}$$

This function is 1 for the minimal binding case, $N = 2$, and approaches zero as $N$ approaches its maximum of 20.

**Protofibril elongation.**   Once a protofibril has formed, it can bind additional fibrin I or II at rate $k_{pg}$ and increase in length. A protofibril can also bind to any oligomer, $f_i$ for $1 < i < 11$, and increase in length accordingly.

$$f_n + f \xrightarrow{k_{pg}} f_n \tag{29}$$

$$f_n + f_b \xrightarrow{k_{pg}} f_n \tag{30}$$

$$f_n + f_i \xrightarrow{k_{pg}^N} f_n. \tag{31}$$

We tracked the number of protofibrils with $f_n$ and continued to track the number of protofibrils associated monomers with $C_{fb}$ and $C_{fn}$. For the linear growth cases, we added to $C_{fb}$ and $C_{fn}$, respectively. But for all other cases we accounted for the fraction of fibrin I and II using $\phi_b$.

$$C_b- = k_{pg} f_b f_n + \phi_b \cdot (i)(k_{pg}^N f_i f_n) \tag{32}$$

$$C_f- = k_{pg} f f_n + (1 - \phi_b) \cdot (i)(k_{pg}^N f_i f_n) \tag{33}$$

$$C_{fb}+ = k_{pg} f_b f_n + \phi_b \cdot (i)(k_{pg}^N f_i f_n) \tag{34}$$

$$C_{fn}+ = k_{pg} f f_n + (1 - \phi_b) \cdot (i)(k_{pg}^N f_i f_n). \tag{35}$$

Additionally, we assumed that $k_{pg}$ decreased as the size of the oligomers binding increased, reflecting that diffusion slows as the size of the oligomers increased, slowing the mass action encounters. Let $k_{pg}^N = k_{pg} \cdot S_{prot}(N)$, where $k_{pg}$ is taken to be the base value for that parameter and N is the size of the oligomer that is binding. Then we scaled the binding rate with the

scaling function:

$$S_{prot}(N) = \frac{11 - N}{10}. \tag{36}$$

so that the scale $S_{prot}$ is 1 for the case that the oligomer was of size one, $N = 1$, and approaches zero as the the oligomer size approached its maximum of $N = 10$.

**Steric inhibition of fibrin binding.** The end-to-end binding of fibrin monomers was also considered, which included fibrin I and II monomers binding directly, oligomers or protofibrils binding to each other, and protofibrils growing in length. It is thought that the presence of $\gamma'$ binding may cause steric inhibition of fibrin-fibrin binding and slow down oligomer and protofibril growth and subsequently slow down polymerization [31]. Consider the probability that for two random fibrin monomers binding, there is a $\gamma'$ chain involved in that binding. Next, consider the fraction of $\gamma$ chains that are $\gamma'$ chains, $f_g$. Call the probability of a fibrin-fibrin binding event occurring in the absence of a $\gamma'$ chain, $P_{gg}$, with exactly one $\gamma'$ chain, $P_{gp}$, and with two $\gamma'$ chains, $P_{pp}$. To allow for steric inhibition due to the $\gamma'$ chain, the rates $k_{pi}$ and $k_{pg}$ were both scaled with the multiplier $S_{bind} \cdot (P_{pp} + P_{pg}) + P_{gg}$, where $S_{bind}$ is a number between 0 and 1. If $S_{bind} = 1$, there would be no inhibition of fibrin monomers by steric interactions due to the $\gamma'$ chain. If $S_{bind} = 0$ then fibrin monomers cannot bind end-to-end because there would be steric hindrance from a $\gamma'$ chain. That steric binding inhibition could potentially effect all fibrin binding, including lateral aggregation, but we assumed for now that it primarily effected the end-to-end binding of fibrin monomers.

**Protofibril lateral aggregation: Fiber formation & fiber growth.** Next are the details for modeling lateral aggregation of protofibrils, where fibrin fibers are formed and grow. Fibrin I is considered necessary for protofibril formation and fibrin II (FpB cleavage) is thought to enhance lateral aggregation in fibrin fibers. We assumed that some fraction of protofibrils can aggregate laterally based on the amounts of oligomer associated with fibrin I to fibrin II. We then refer to the polymerizable fraction, $\phi_p$, where

$$\phi_p = \frac{C_{fn} + S_{agg} \cdot C_{fb}}{C_{fn} + C_{fb}} \tag{37}$$

We considered the fraction of protofibrils that were able to aggregate laterally into fibrin fibers as $f_n^p$ where $f_n^p = \phi_p \cdot f_n$. $S_{agg}$ is a parameter that lets us interpolate between the two extremes of FpB effects on lateral protofibril aggregation into fibrin fibers. When $S_{agg} = 0$, FpB completely inhibits lateral aggregation of protofibrils. When $S_{agg} = 1$, FpB has no effect on the lateral aggregation of protofibrils, and this reduced the model to a case where only FpA cleavage affects fiber formation. If $0 < S_{agg} < 1$, then there was partial inhibition of protofibril lateral aggregation into fibrin fibers by the presence of FpB, ranging between the two extremes.

Protofibrils were assumed to aggregate laterally in two different ways. They can bind to each other at a rate, $k_{fi}$, forming a new fiber, $f_r$ or they can bind laterally to an existing fiber at a rate, $k_{fg}$, increasing the width of that fiber:

$$f_n^p + f_n^p \xrightarrow{k_{fi}} f_r \tag{38}$$

$$f_r + f_n^p \xrightarrow{k_{fg}} f_r. \tag{39}$$

We tracked the total concentration of fibrin monomers associated with fibers as $C_{fr}$, subtracting from $C_{fb}$ and $C_{fn}$ and adding to $C_{fr}$ when appropriate. We tracked the fibrin monomers

moving from protofibrils to fibers as:

$$C_{fb}- = S_{agg}(2k_{fi}f_n^p C_{fb} + k_{fg}f_r C_{fb}) \tag{40}$$

$$C_{fn}- = 2k_{fi}f_n^p C_{fn} + k_{fg}f_r C_{fn} \tag{41}$$

$$C_{fr}+ = S_{agg}(2k_{fi}f_n^p C_{fb} + k_{fg}f_r C_{fb}) + 2k_{fi}f_n^p C_{fn} + k_{fg}f_r C_{fn}. \tag{42}$$

**Bivalent binding, FpB cleavage on oligomers & protofibrils.**   The bivalent binding of thrombin to oligomers and protofibrils allowed for the coupling of the polymerization and thrombin cleavage of fibrinopeptides components of the model described above. We incorporated the bivalent binding scheme and estimated binding rates from our previous work [24]. Here, we discuss how to account for $\gamma'$ binding sites within polymerized oligomers, protofibrils, and fibers. We denote $E_1$ and $G_1$ as E-domain and $\gamma'$-associated binding sites that were not in proximity to one another, respectively, and thrombin bound to these sites is $BE_1$ and $BG_1$. Note that bivalent binding is not a possible transition from these states. Similarly, $E$ and $G$ are the E-domain and $\gamma'$ associated binding sites that are in proximity to each other, with $BE$ and $BG$ as the corresponding thrombin-bound species that can transition into bivalently bound thrombin, $B$. It is possible that $E_1$ and $G_1$ and the associated bound thrombin may transition to become within proximity of one another as fibrin monomers line up length wise during aggregation. These changes were monitored by tracking the amount of fibrin(ogen) that was polymerized, be it in oligomers, protofibrils, or fibers, and the amount of fibrin(ogen) monomers, i.e., free fibrinogen, fibrin I, and fibrin II.

Since we allowed for fibrin I and fibrin II to aggregate, we further assumed that thrombin bound to the E-domain, $E_1$, $E$, and $B$ could cleave FpB on fibrin I to convert it to fibrin II within the polymerized/aggregated oligomers and protofibrils, and fibers. It is thought that E-domain bound thombin cleaves FpB, so we assumed that all forms of this thrombin, including the bivalently bound form, could cleave FpB but might do so at different rates. This is an assumption we made based on the allosteric behavior of thrombin. To allow for the different cleavage rates within the model, we introduced three scales $S_\alpha$, $S_\beta$, and $S_\gamma$, that multiply the catalytic rate constant for FpB cleavage by thrombin bound to an E-domain far from a $\gamma'$ ($BE_1$), an E-domain near a $\gamma'$ ($BE$), and thrombin that is bivalently bound, ($B$), respectively. Another consideration was whether thrombin dissociated from its fibrin(ogen) binding sites upon cleavage of FpB. We assumed that thrombin bound only to an E-domain site would immediately dissociate and that some fraction $R_{biv}$ of bivalently bound thrombin would dissociate from the fibrin and become free thrombin, and $1$-$R_{biv}$ would stay bound to the $\gamma'$ but dissociate from the E-domain. This can all be seen in the reactions:

$$BE_1 \xrightarrow{S_\alpha k_{cat,b}} E_1 + T \tag{43}$$

$$BE \xrightarrow{S_\beta k_{cat,b}} E + T \tag{44}$$

$$B \xrightarrow{S_\gamma R_{biv} k_{cat,b}} E + G + T \tag{45}$$

$$B \xrightarrow{S_\gamma(1-R_{biv})k_{cat,b}} E + BG \tag{46}$$

The overall cleavage of FpB within polymerized fibrin(ogen) is then:

$$F_{FpB}(BE_1, BE, BG) = k_{b,cat}(S_\alpha BE_1 + S_\beta BE + S_\gamma B). \tag{47}$$

**Table 2. Parameters for the polymerization model.**

| Parameter | Description | Baseline Value | Estimated Values |
|---|---|---|---|
| $k_{fi}$ | Rate for monomer-monomer binding (Oligomer formation) | $6.022e\text{-}4\ (\mu Ms)^{-1}$ | $0.0091\ (\mu Ms)^{-1}$ |
| $k_{fg}$ | Rate of protofibril growth (elongation) | $12.00\ (\mu Ms)^{-1}$ | $32.607\ (\mu Ms)^{-1}$ |
| $k_{pi}$ | Rate of protofibril-protofibril binding (fiber iniation) | $2.409\ (\mu Ms)^{-1}$ | $2.953\ (\mu Ms)^{-1}$ |
| $k_{pg}$ | Rate of fiber protofibril binding (lateral fiber growth) | $60.22\ (\mu Ms)^{-1}$ | $0.4581\ (\mu Ms)^{-1}$ |
| $S_{agg}$ | Efficiency of Fibrin II to laterally aggregate | 0.5 | 0.0921 |
| $S_{\alpha}$ | Scale for FpB cleavage from $BE_1$ | 1.0 | 1.0164 |
| $S_{\beta}$ | Scale for FpB cleavage from $BE$ | 1.0 | 1.0164 |
| $S_{\gamma}$ | Scale for FpB cleavage from $B$ | 1.0 | 0.9900 |
| $R_{biv}$ | Fraction thrombin released upon FpB cleavage | 0 | 0.7126 |
| $S_{bind}$ | Rate of steric inhibition on fibrin-fibrin binding | 1.0 | 0.2524 |

This function was multiplied by the fraction of binding sites occupied by thrombin to obtain the overall rate of FpB cleavage, and then used to update the corresponding conversions from $C_b$ to $C_f$ and $C_{fb}$ to $C_{fn}$.

## Parameter estimation

The final ODE polymerization model has 9 unknown parameters, listed and described in Table 2. The four rates $k_{pi}$, $k_{pg}$, $k_{fg}$, and $k_{fi}$ are assigned but largely uncertain values taken from the previous study of Weisel and Nagaswami [11]. Thus, we chose to include them in our parameter estimation framework. The efficiency at which fibrin I can laterally aggregate compared to fibrin II is $S_{agg}$. We primarily considered the case where $0 < S_{agg} \leq 1$, which assumes that fibrin II is always able to laterally aggregate. However, we did consider the case where $S_{agg} > 1$ as a control to make certain that we did not constrain out parameter set by our assumptions. Three parameters control the relative effectiveness of thrombin, bound in three different configurations, to cleave FpB: $S_{\alpha}$ is the scale for E-domain bound thrombin cleaving FpB, $S_{\beta}$ is the scale for E-domain bound thrombin in proximity to a $\gamma'$ chain cleaving FpB, and $S_{\gamma}$ is the scale bivalently bound thrombin can cleave FpB. When the parameters $S_{\alpha}$, $S_{\beta}$, and $S_{\gamma}$ are greater than one, they represent catalytic enhancement above baseline whereas when they are between 0 and 1, they represent a reduction. However, we assumed that all E-domain bound thrombin cleaves FpB at the same rate so that $S_{\alpha} = S_{\beta}$. We also considered $R_{biv}$ as a parameter to estimate.

We started with values from Weisel and Nagaswami [11] for the polymerization rates. We let $S_{agg} = .5$ be our baseline initial guess, representing fibrin I lateral aggregation at 50% the rate of Fibrin II. We set $S_{\alpha} = S_{\beta}$, which assumes that all E-domain, fibrin bound thrombin cleaves FpB at the same rate. We allowed $S_{\alpha} = S_{\beta} = S_{\gamma}$ but to vary between zero and infinity, with a starting guess of 1. We considered $R_{biv}$ and $S_{bind}$ to vary between 0 and 1. The initial value for $R_{biv}$ was 0, and the initial value for $S_{bind}$ was 1. We then performed parameter estimation on all 9 of these parameters. They were all unknown or there was little to no evidence of a best guess without further validation or verification [11, 31, 32]. Our algorithm for parameter estimation is as follows:

1. Use MATLAB's fmincon function to estimate the 9 unknown parameters; check the minimized sum of squares between the model output for clot-time and the data from Kim et al. [38]

   - Use the baseline values listed in Table 2 as the initial values

- Assign a random starting point in parameter space by adding a normal random number (with small variance) to the initial guesses

- Verify that all parameters were non-negative

- Run MATLABS's fmincon for 800 of these random starting points, find local minimums in parameter space while avoiding getting stuck with a local minimum and not a global minimum

- Use qualitative results of fibrin formation to further refine our selection.

From the results of Wesiel and Nagaswami [11] and Gersh et al. [31] we defined two additional qualitative selection criteria:

1. An increase in thrombin must decrease protofibril number,

2. An increase in $\gamma'$ binding sites must decrease protofibril number.

These two criteria were used to discriminate between multiple possible solutions from the parameter estimation process. Additionally, from the results of Domingues et al. [32] we know that as thrombin in increased and fibrin(ogen) is fixed that $\gamma_A/\gamma'$ fibrinogen should produce smaller, less dense fibers with a slightly larger distance between packed protofibrils at lower thrombin concentrations.

**Model construction.** The polymerization model consists of a system of ordinary differential equations, was assembled using the law of mass action, and the equations were solved in MATLAB using the built-in ODE solver ode15s. The model equations in their entirety are detailed in S1 Appendix; here we will just look at an example of a species of oligomers, $f_i$, to give some intuition about the types of terms included in the model equations. We considered the formation and growth of the oligomer, $f_i$, as well as the depletion, and $f_i$ also contributes to the formation or growth of other oligomer and protofibrils. Thus, the equation is as follows:

$$
\frac{d[f_i]}{dt} = \underbrace{k_{pi}[f_{i-1}]([f]+[f_b])}_{\text{Oligomer formation by a single fibrin monomer}} + \underbrace{k_{pi}\sum_{\substack{j+k=i \\ 2\le j\le k}}[f_j][f_k]}_{\text{Oligomer formation from the combination of two smaller oligomers}}
$$

$$
- \underbrace{k_{pi}[f_i]([f]+[f_b])}_{\text{Oligomer growth by a single fibrin monomer}} - \underbrace{k_{pi}[f_i]\sum_{j=1}^{10}[f_j]}_{\text{Oligomer$-$oligomer binding, forming both oligomers and protofibrils}} \tag{48}
$$

$$
- \underbrace{k_{pg}[f_i][f_n]}_{\text{Oligomer$-$protofibril binding, which increases protofibril length}} .
$$

**Calculating trapped thrombin.** Consider the cross-section of a fibrin fiber of diameter $d$, with some number of protofibrils, $N_{pf}$, arranged roughly in concentric layers throughout. We assumed that free thrombin is accessible to the outermost two layers of protofibrils only and that any thrombin that is within the interior core was deemed 'trapped'. The number of protofibrils that were assumed to be inaccessible to free thrombin is $N_I$, accounted for the protofibrils in the core of the cross-section; these numbers were computed using $d$, $N_{pf}$ and a circle packing algorithm, as in our previous work [24]. Let $I(N_{pf}) = \frac{N_I}{N_{pf}}$ be the fraction of inaccessible protofibrils in a fiber with $N_{pf}$ protofibrils. $I(0)$ is defined to be zero to avoid division-by-zero. To calculate the concentration of bound thrombin on the fibrin fibers themselves, one must think of the fraction of E-domain-bound thrombin, $\gamma'$-bound thrombin, and bivalently-bound

thrombin on the fibrin fibers and then consider the total number of binding sites in the system (for this example, we use wild-type fibrin(ogen):

$$N_{E1} = 2f + 1.7\left(\left(\sum_{i=2}^{10} if_i\right) + C_{fb} + C_{fn} + C_{fr}\right) \tag{49}$$

$$N_E = 0.3\left(C_f + \left(\sum_{i=2}^{10} if_i\right) + C_{fb} + C_{fn} + C_{fr}\right) \tag{50}$$

$$N_{G1} = 0.3(f_{ab} + f_b + f + Ef_{ab} + Ef_b) \tag{51}$$

$$N_G = 0.3(C_f + \left(\sum_{i=2}^{10} if_i\right) + C_{fb} + C_{fn} + C_{fr}) \tag{52}$$

Then the fraction of those binding sites that are part of a fibrin fiber are:

$$\phi_{BE1} = \frac{1.7 \cdot C_{fr}}{N_{E1}} \tag{53}$$

$$\phi_{BE} = \frac{0.3 \cdot C_{fr}}{N_E} \tag{54}$$

$$\phi_{BG1} = \frac{0 \cdot C_{fr}}{N_{G1}} = 0 \tag{55}$$

$$\phi_{BG} = \frac{0.3 \cdot C_{fr}}{N_G} \tag{56}$$

The total concentration of bound thrombin would then be

$$T^{tot}_{bound} = BE1 + BE + BG + BG1 + B, \tag{57}$$

and the total amount of thrombin bound to fibrin fibers is

$$T^{tot}_{fiber} = BE1 \cdot \phi_{BE1} + BE \cdot \phi_{BE} + BG1 \cdot 0 + BG \cdot \phi_{BG} + B \cdot \phi_{BG}. \tag{58}$$

The fraction of fiber-bound thrombin is

$$\phi_{Tfiber} = \frac{T^{tot}_{fiber}}{T^{tot}_{bound}}. \tag{59}$$

Then, $I(N_{pf})$ is interpolated over the number of protofibrils per fiber, $m$, so that the fraction of trapped thrombin is

$$\phi_{trap} = I(m) \cdot \phi_{Tfiber} \tag{60}$$

## Results

### Stochastic binding model

The stochastic binding (SB) model was used to estimate the amount of time that a thrombin molecule will stay bound to polymerized $\gamma_A/\gamma_A$ fibrin or $\gamma_A/\gamma'$ fibrin within monomer-monomer junctions, while being subjected to a wide range of removal rates. We considered removal

rates over seven orders of magnitude: $10^{-2}s^{-1}$ to $10^5s^{-1}$ to represent a variety of removal scenarios, e.g., diffusion and/or flow at arterial, venous, and pathological shear rates. We will refer to two binding scenarios: thrombin binding to a junction within $\gamma_A/\gamma_A$ fibrin (AA junction) and a junction within $\gamma_A/\gamma'$ fibrin (AP junction), refer to Fig 1 for the various species included at each junction. The corresponding times were computed for wild-type fibrin by using a weighted average of the times estimated for AA and AP junctions (85:15 ratio of AA to AP). These three cases give some intuition about extreme cases of binding as well as the physiologically relevant case.

The average time that thrombin spent in any state before removal was tracked for nine removal rates values, $k_r$, for both the AA and AP junctions. In Fig 3A, we report the mean time to removal for AA (blue curves), AP (black curves), and the weighted average for wild-type (magenta curves). The times to removal included thrombin in all possible states, including free, except the removed state. In all cases, thrombin was removed more quickly as the removal rate increased, which was expected. Thrombin stayed within an AP junction for about 1.5 hours at the lowest removal rate. The mean time in that case approached a minimum time of approximately 30 seconds at the highest removal rate. Thrombin spent about 4 minutes in an AA junction for the lowest removal rate and the mean time approached a minimum time of 0.3 seconds at the highest removal rate. The weighted average of the two cases, the wild-type case, led to thrombin with a mean time to removal of 15 minutes for the smallest removal rate and 4.5 seconds at the highest rate. These results demonstrate that thrombin can potentially stay within any of these junctions, even when subjected to active removal, for timescales on the order of fibrin polymerization.

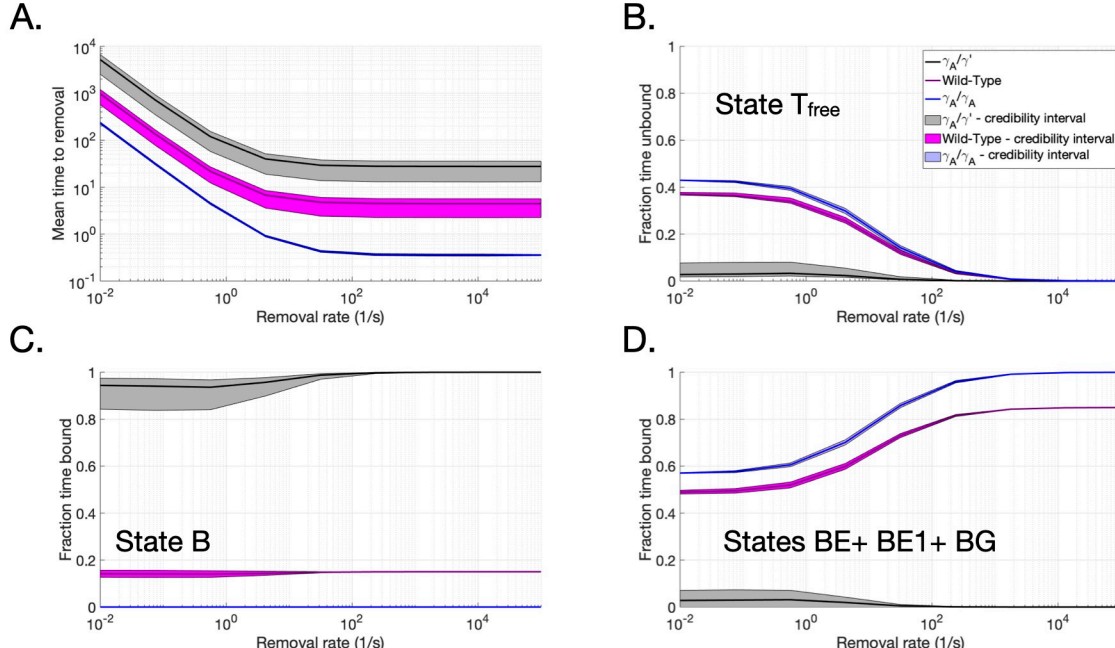

**Fig 3. Time of thrombin in free and bound states within AA, AP, and wild-type junctions.** Reported means and credibility intervals for quantities of interest from the stochastic binding model for various junctions and removal rates. Free and bound states of single thrombin molecules were tracked within AA junctions ($\gamma_A/\gamma_A$ fibrin), AP junctions ($\gamma_A/\gamma'$ fibrin). Wild type fibrin was represented as a weighted average of these two cases. Thrombin was removed from the system at removal rates, $k_r$. A) Mean time to removal starting from bound states, to either an E-domain or bivalently, fraction of mean time to removal spent (B) susceptible to removal, (C) bound to high-affinity bivalent sites, and (D) bound to low-affinity sites.

The mean times to removal were further parsed to examine the fraction of time spent free and susceptible to removal (Fig 3B), bound to low-affinity binding sites (Fig 3C), and bound to high-affinity binding sites (Fig 3D) in all types of junctions. At the lowest removal rates, thrombin in an AP junction spent significantly less time susceptible to removal than thrombin in an AA or wild-type junction, with thrombin in AA and wild-type junctions spending about 40% of time in the free state compared to thrombin in the AP junction spending less than 10% of time in that state. As the removal rate increased, the fraction of time susceptible to removal decreased to zero for all cases, indicating that the rest of the time to removal was spent in a bound state. Fig 3C and 3D show the corresponding times spent bound to high-affinity associated binding sites and low-affinity associated binding sites, respectively. Thrombin within an AP junction spent between 94–100% of its time bound to high-affinity associated binding sites and between 0–2% of its time bound to low-affinity associated binding sites. Thrombin within an AA junction spent none of its time bound to high-affinity associated binding sites since none exist there and between 57–99% of its time bound to low-affinity associated binding sites. Thrombin in a wild-type junction was estimated to have spent between 45–81% of its time bound to low affinity binding sites and a near constant 18% of its time bound to the high-affinity associated bindings sites. In summary, the stochastic binding model showed that thrombin within an AP junction spent significantly more time bound than thrombin within an AA junction. In both cases, thrombin spent a non-zero amount of time bound and, when possible, spent most of its time bound to the high-affinity bivalent binding sites.

## Polymerization model results

The results below are from the integrated polymerization model that includes thrombin binding with fibrinogen and fibrin, fibrin formation and subsequent polymerization. Model outputs of interest are the rates and total fractions of fibrinopeptides cleaved during polymerization, the number of protofibrils per fiber, $m$, which was used as a proxy for fiber thickness, and the amount of trapped thrombin within forming fibrin fibers. This section begins with a presentation of results using previously published 'baseline' parameter values [11, 24] and/or our best guesses of what these parameter values should be. We made an additional assumption that FpB was cleaved at the same rate by all E-domain-bound thrombin species. The results under these conditions were not in line with turbidity experiments and thus illustrated the need for further assumptions within the model and an exploration of different parameter values. After such a calibration, the model qualitatively reproduced experimental observations from disparate experimental data sets, gave estimates of trapped thrombin quantities during polymerization, and provided potential underlying mechanisms for these observations.

**Model with baseline parameter values and new assumptions.**   We first performed simulations with parameters at baseline values, as listed in Table 2. One goal of this study was to better understand experimental results showing that increased $\gamma'$ fibrin(ogen) leads to thinner fibrin fibers [31, 32], a behavior observed with increasing thrombin concentrations. These results are nonintuitive if $\gamma'$ fibrin(ogen) acts purely as a high-affinity thrombin sink. We considered the output for protofibril number, as thrombin was varied between 0.1 and 100 nM (Fig 4A) and when the ratio of $\gamma'$-chain to E domains was varied (Fig 4B). The model predicted the correct qualitative behavior as thrombin was increased, i.e., shorter lag time (time to half the maximal number of protofibrils) and thinner fibers. However, the changes in predicted protofibril numbers as the ratio of $\gamma'$-chain to E domains increased were in the wrong order (i.e., purple curve lower than blue curve). Additionally, the lag times in both cases were at least an order of magnitude larger than those seen experimentally [11, 31, 37]. Parameter estimation

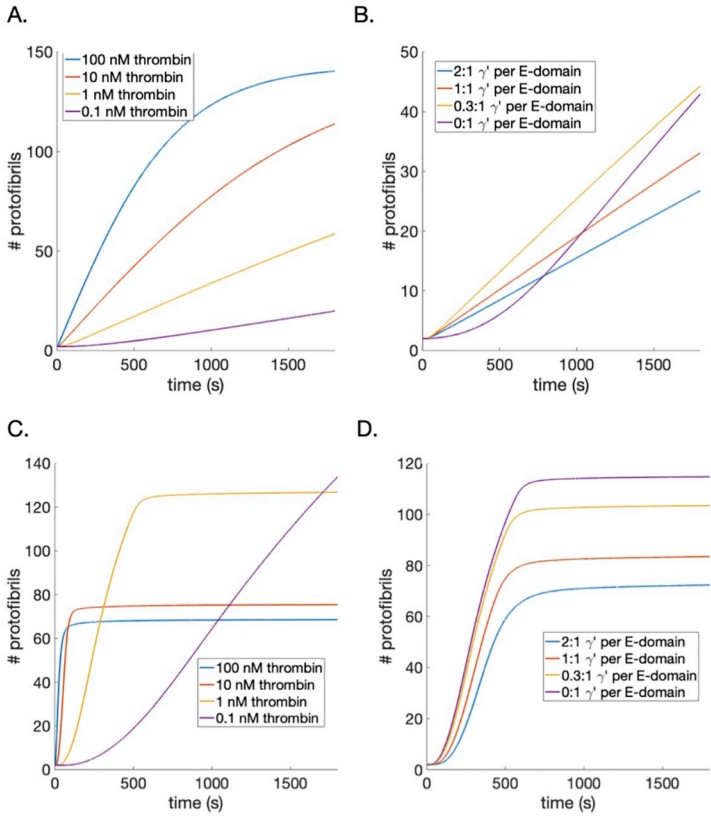

**Fig 4. Number of protofibrils per fiber during polymerization.** Model output for protofibril number as a function of thrombin concentration and ratio of γ':E-domains. Varying thrombin concentration, 0.1 *nM*—purple, 1.0 *nM*—yellow, 10 *nM*—red, 100 *nM*—blue, using previously published parameters. B) The ratio γ' binding sites to fibrin (ogen) monomers (0:1—purple, 0.3:1—yellow, 1:1—red, and 2:1—blue) is varied and model uses previously-published parameters. Varied thrombin from 0.1 *nM*—purple, 1.0 *nM*—yellow, 10 *nM*—red,100 *nM*—blue, (C) and varied ratio of γ' binding sites to fibrin(ogen) monomers, from 0:1—purple, 0.3:1—yellow, 1:1—red, and 2:1—blue, (D) from the model evaluated with best-fit parameters.

attempts alone, using the methods described above, were not able to achieve the correct ordering of the curves.

**Parameter estimation.**    Our model included new mechanisms that affect polymerization such as steric inhibition, thrombin-fibrin interactions, and the formation of fibrin monomers fibrin II. Thus, it was not expected that the parameters governing oligomer, protofibril, and fiber formation, $k_{fi}$, $k_{fg}$, etc., used in the previously published polymerization model [11] would necessarily be appropriate in this setting. The new model parameter, $S_{bind}$ was added to include steric inhibition and affect the binding 'strength' of γ' fibrin as compared to $γ_A$ fibrin. We reassessed the assumption that FpB could be cleaved in the: (i) E-domain far from a γ' chain, (ii) E-domain near a γ' chain, and (iii) E-domain with bivalently bound thrombin to a nearby γ' chain. The original assumption was that thrombin dissociated from the E-domain, in both cases (i) and (ii), upon enzymatic cleavage of FpB but the bivalently bound thrombin was assumed to stay bivalently bound after cleavage. It seems possible that bivalently bound thrombin, upon enzymatic cleavage of FpB, could also unbind from the E-domain and stay bound to the γ' site or even dissociate completely from the fibrin and become free thrombin *T*. We incorporated these possibilities into the model with the parameter, $R_{biv}$, the fraction of bivalently bound thrombin that dissociates into free thrombin upon enzymatic cleavage of FpB,

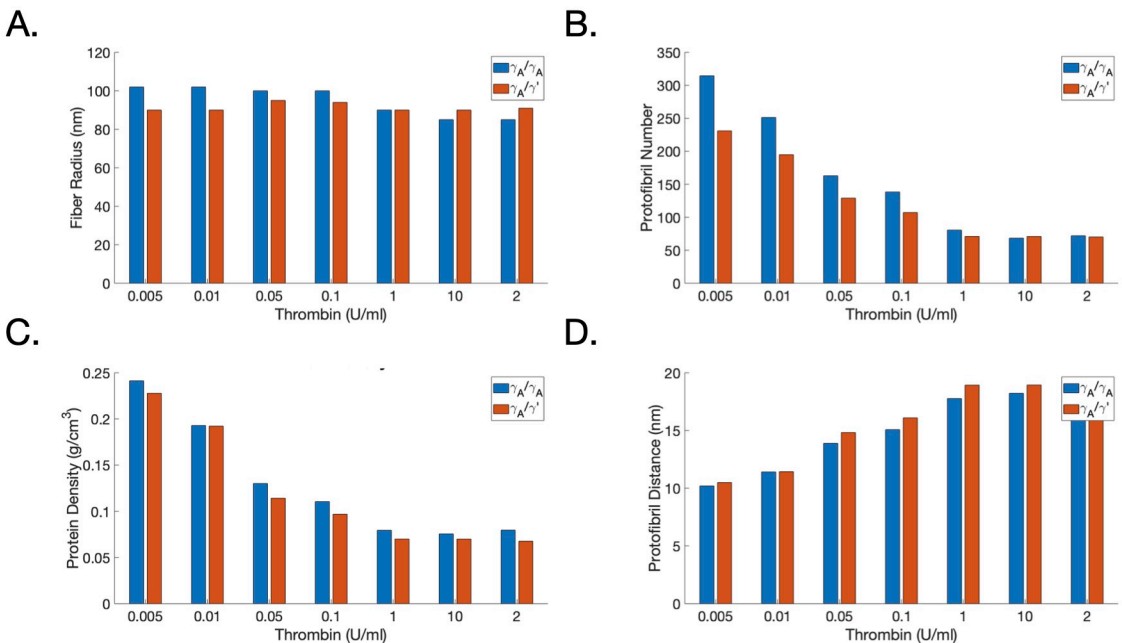

**Fig 5. Box plot of parameter estimates and relative clot time vs. fibrinogen.** Left: Box and whisker plot for 300 parameter sets with the smallest least squares errors from the parameter estimation process. The parameter values are shown in log-scale and the units are those in Table 2. Right: The ratio of clot times showing the relative clot time between $\gamma_A/\gamma'$ to $\gamma_A/\gamma_A$ fibrinogen. Clot time being defined as the time to half-maximal turbidity measurement. The model output (solid line) is below the experimental data from Kim et al. [38] (dots with error bars).

where $1 - R_{biv}$ is the fraction that released from the E-domain but stayed bound to the $\gamma'$ chain. Nine parameters were estimated using the clot time data from Kim et al. [38] and the numerical techniques described in the methods section; these parameters and estimated values are listed in Table 2. The distribution of the estimated parameters and their fit to the data are shown in Fig 5. The set with the minimal least squares error is provided in Table 2. This set of parameters was then used to recreate the scenarios considered in Fig 4A and 4B, with new results shown in Fig 4C and 4D. In this new set of simulations, the correct behavior was observed: when the type and concentration of fibrin(ogen) was fixed ($3.29\mu M$) and thrombin was increased, there was a decrease in protofibril number and the lag times were all within experimentally observed times (Fig 4C), and the ratio of $\gamma'$-chains to E domains was increased, there was a clear decrease in protofibril number as observed experimentally (Fig 4D).

## Qualitative agreement with experimental studies

The model behavior with the best-fit parameters was qualitatively consistent with a few different experimental data sets found in the literature that varied thrombin, fibrin(ogen) type and concentration [11], measured cleavage of fibrinopeptides [31] and were run for both long and short periods of time investigating steric interactions [32]. The results in the next few sections will address these cases and additional confirmation of the hypothesis from our previous work regarding thrombin trapping during polymerization.

**Thrombin, fibrin(ogen) type, and structure.**   The data from Domingues et al. [32] included structural measurements such as fiber radius, protein density, and distance between protofibrils, as well as protofibril number, each as a function of thrombin concentration and fibrin(ogen) type. We performed simulations of the experiments in that study and show the

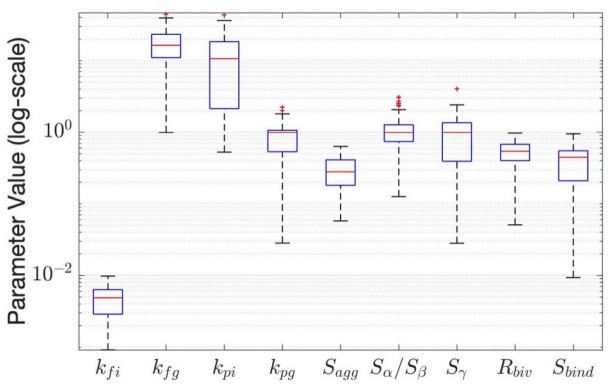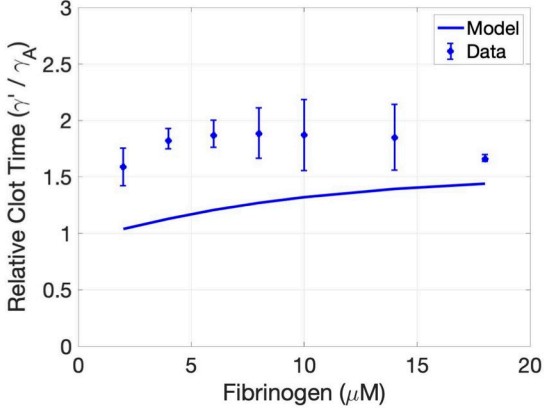

**Fig 6. Protofibril number and structural quantities after long periods of time: Comparing model output with experiments.** Shown are the experimentally measured fiber radii [32] in (A). Model outputs as a function of both the thrombin concentration and $\gamma'$:E-domain ratio: (B) protofibril number, (C) protein density and (D) distance between protofibrils. The quantities in C,D are computed by combining the measurements in A with a post-processing of the model output in B.

results in Fig 6. The model behavior captured the behavior reported in the Domingues et al. study. In particular the protofibril number for $\gamma_A/\gamma_A$ was indeed higher than that of $\gamma_A/\gamma'$ at thrombin concentrations strictly less than 1 U/ml (10 nM) and above that, the protofibril numbers were almost the same Fig 6B. We found similar behavior in the post-processed protein densities and the distances between protifibrils, shown in Fig 6C and 6D, estimated using the measured width of the fibers (Fig 6A) from Domingues et al. [32].

**Bivalent binding and FpB cleavage.** To better understand how bivalent binding of thrombin with fibrin affected fibrinopeptide cleavage during polymerization, we performed simulations with the best fit parameters and monitored the FpA and FpB cleavage for three different ratios of $\gamma'$:E-domains: 2:1 ($\gamma'/\gamma'$), 1:1 ($\gamma'/\gamma_A$), and 0:1 ($\gamma_A/\gamma_A$). For all three types of fibrin(ogen), the rate of FpA cleavage was nearly identical, with all FpA cleaved after 40 minutes (dashed lines in Fig 7A). It makes sense that these rates would look the same since FpA cleavage must occur before aggregation and bivalent binding can only occur within polymerized forms of fibrin. For FpB cleavage there was a distinct decrease in the rate of cleavage as the ratio was decreased from 2:1, 1:1, to 0:1, with all FpB cleaved by 50 (blue line), 90 (red line), and 125 (yellow line) minutes, respectively (Fig 7A). These results are in line with experimental observations [31].

Since our model explicitly considered thrombin fibrin interactions, the distribution of thrombin in its various states during fibrinopeptide cleavage could also be observed. Free or bound states of thrombin clearly showed two phases during polymerization. During the first phase, free thrombin was available to cleave and bind in all cases until all FpB's were depleted, after which it entered a second phase, shifting rapidly from being free to being bound, seen as sudden drops in the curves in Fig 7B between about 40 and 80 minutes. With nonzero ratios of $\gamma'$:E-domain, almost all of the free thrombin became bound during the second phase of polymerization (blue and orange curves) whereas with just $\gamma_A$ and low-affinity binding, about 80% of the thrombin remained free in the second phase (yellow curve). The bound thrombin was distributed between the low-affinity binding sites, $B_{E1}$ and $B_G$, shown in Fig 7C, and those associated with bivalent binding, $B_E$, $B_G$, and $B$, shown in Fig 7D. In cases where thrombin could bind bivalently, almost all of the thrombin was bound in that manner compared to what was bound to the lower affinity sites. In summary, these data show that free thrombin is 'recycled' to continue further cleavage of fibrinopeptides during the early stage of polymerization,

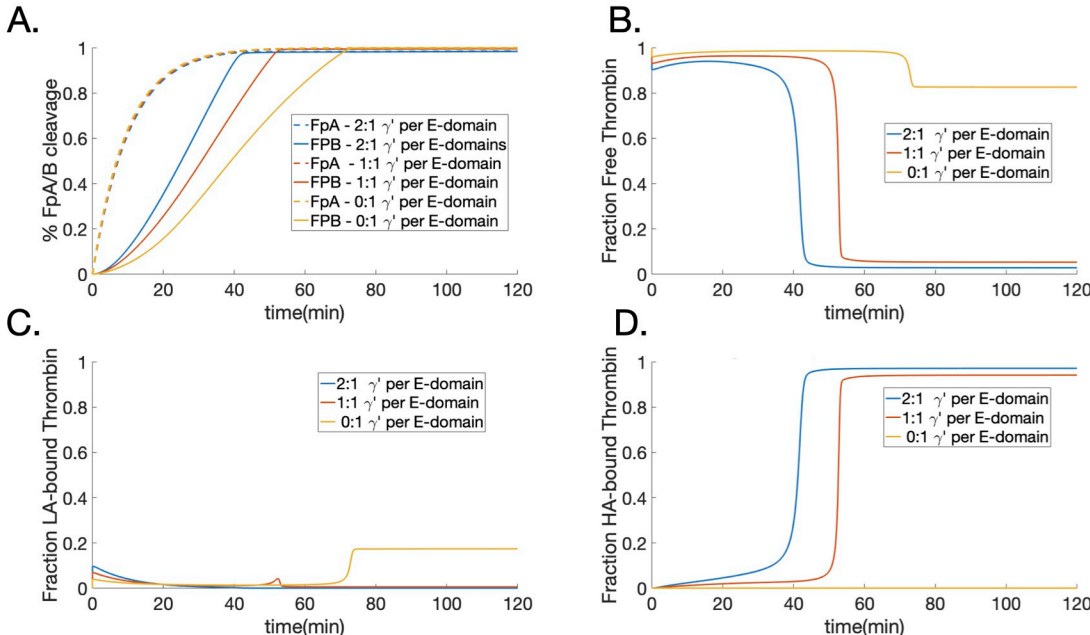

**Fig 7. Fibrinopeptide cleavage and thrombin binding.** Three typed of fibrin(ogen) are considered: $\gamma'/\gamma'$ fibrinogen (blue), $\gamma_A/\gamma'$ fibrinogen (red), and $\gamma_A/\gamma_A$ fibrinogen (yellow). Shown are time courses of (A) the percent of FpA (dashed lines) and FpB (solid lines) cleaved due in the presence of 0.1 nM thrombin and 0.1 mg/ml fibrinogen, (B) the percent of free thrombin, (C) the percent of thrombin bound to low-affinity binding sites (those not associated with bivalent binding), $BE1$ and $BG$, and (D) the percent of thrombin bound to high-affinity, bivalent sites: $BE$, $BG$, and $B$.

but after depletion of fibrinopeptides, thrombin becomes bivalently bound and strongly sequestered during the later stage.

**Estimates of trapped thrombin during polymerization.** Since the model could be used to track the amount of thrombin bound to monomers, oligomers, protofibrils, and fibers, it could also be used to estimate how much thrombin is trapped within fibers during polymerization. We considered the same two cases as depicted in Fig 4: varying thrombin concentration with wild-type fibrin(ogen) and varying the ratio of $\gamma'$ binding sites to fibrin(ogen) monomers. The dynamic time courses of the thrombin trapped within fibers, computed as a percent of the total bound thrombin (see Methods), for these two cases are shown in Fig 8A and 8B, respectively. The percent of trapped thrombin intuitively tracks with the protofibril number when comparing to Fig 4C and 4D. In both cases, the estimated percent of bound thrombin trapped within the fibers was in a range of 30–45%. Interestingly, this is very close to the range of 23–39% percentage we previously estimated from a reaction-diffusion model of thrombin dissociation from preformed fibrin clots [24]. The model suggests that a significant amount of thrombin could indeed become trapped within fibers during the timescale of polymerization.

## Discussion

### Stochastic binding model and the potential for thrombin trapping

Results from our stochastic binding model showed that thrombin spent significantly more time within an AP junction than it did within an AA junction before being removed, over a wide range of removal rates. This was due to the fact that thrombin spent a larger fraction of time in an unbound state and was thus susceptible to removal within an AA junction as compared to an AP junction. Additionally, thrombin in an AP junction was primarily in the

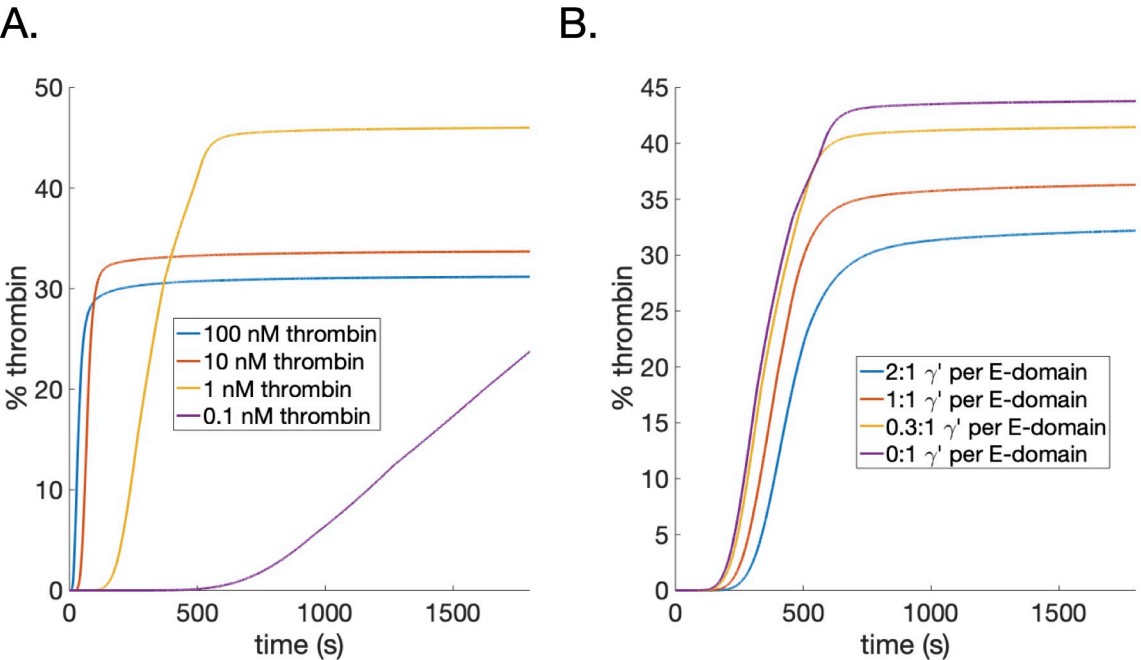

**Fig 8.** Estimates of percent of bound thrombin that is trapped within fibers as a function of increasing: A) thrombin (0.1 $nM$—purple, 1.0 $nM$—yellow, 10 $nM$—red,100 $nM$—blue) and B) the ratio $\gamma'$ binding sites to fibrin(ogen) monomers (0:1—purple, 0.3:1—yellow, 1:1—red, and 2:1—blue). As thrombin is increased, percent of thrombin trapped decreases following the protofibril number, with the exception of the 0.1 $nM$ where enough time has not elapsed for it to surpass the yellow curve. As the $\gamma'$ ratio increases, the trapped thrombin monotonically increases.

bivalently bound state and was thus shielded from removal; likely because thrombin in that case would have to go through two sequential unbinding steps before becoming susceptible to removal. Thrombin bound to a low-affinity E-domain needs only one transition to become susceptible to removal. Times to removal of thrombin near a wild-type junction were estimated as the weighted average of the times in an AA and AP junction, spending about 15–20% of the time in a bivalently bound state, even at very high removal rates.

These results suggest that thrombin may spend a significant fraction of time bound to or very near (and thus likely to bind) fibrin before being removed by diffusion or flow. Even under the highest removal rates, thrombin near fibrin to which it can bivalently bind could spend upward of 30 seconds in a bivalent fibrin junction before being removed. Experimental findings show significant fiber formation within minutes of plasma exposure to tissue factor [41] and fibrinogen exposure to thrombin [38]. Therefore, the timescale estimated here suggests it is quite reasonable for thrombin to stay near or bound to a junction where polymerization reactions are occurring and therefore could become trapped within junctions as fibers form and grow. As protofibrils laterally aggregate and fibers continue to form, removal rates of thrombin in the core of the fibers likely decrease due to physical shielding of the additional protofibril layers, and thus the times to removal would further increase. The removal rates and times to removal may vary significantly between thick and thin fibrin fibers but our results suggested this trapping to be quite robust over even the highest removal rates. This suggestion is also in line with previous findings that thrombin is trapped within clots [23] and the hypothesis that thrombin is trapped within fibers [24].

Chen and colleagues recently reported an estimated half-life for thrombin in a fibrin clot exposed to flow to be about 1.1 hrs [42]. To compare our results here to that reported value,

we considered a similar sized fibrin clot of $10 \times 10 \times 15 \ \mu m$, with thrombin removal at venous shear rates. Thrombin can, at most, interact with a binding junction every 22.5 nm (half the length of fibrin monomers). This means that thrombin has a maximum of 444.45 opportunities to bind along a 10 $\mu$m in any direction within the clot. The mean time bound to a wild-type junction from our stochastic model is 60–1000 seconds over the range of removal rates we investigated. Multiplying this time range by the maximal number of binding events gives a total time bound range between 26,667 and 444,450 seconds, or between 7.4 and 123.46 hours. We will now ignore the time between binding events; the time taken for thrombin to diffuse 10 $\mu$m is on the order of $10^{-5}$ seconds, using a diffusion coefficient of $1.42 \times 10^{-8}$ cm$^2$/s. Next, we assumed the volume fraction of fibrin to be about 0.02, using concentrations of 3 mg/ml fibrin from data on the hydraulic permeability of fibrin clots [43]; note that this is a higher concentration than that used by Chen and colleagues, which was in a range if 0.088 to 0.551 mg/ml [42]. Thus, with a volume fraction of 0.02, thrombin spends 2% of the time estimated above, which comes to a range of 0.15 and 2.47 hours, in the clot before removal. Multiplying this range by ln2, assuming first order removal, leads to a range in half-life of 0.1 to 1.71 hours, in agreement with the previous estimate of thrombin removal under flow.

## Polymerization model

Building on the pioneering work of Weisel and Nagaswami [11], and Naski and Shafer [9], we developed a continuum model of fibrin polymerization that considered $\gamma'$ fibrinogen and explicitly tracked thrombin-fibrin(ogen) interactions, including bivalent binding of thrombin with fibrin. This model extended the previous work by allowing for both fibrin I and fibrin II formation, and all possible combinatoric interactions between oligomers and protofibrils. It also included distinct lateral aggregation rates for fibrin I and fibrin II and incorporated inhibition of end-to-end fibrin binding in the presence of $\gamma'$ chains due to steric interactions. The explicit inclusion of thrombin and fibrin(ogen) binding not only allowed for tracking of fibrinopeptide cleavage on monomers, protofibrils, oligomers, and fibers, but also allowed for the possibility that thrombin could potentially cleave FpB at different rates depending on its bound state and location. This assumption can be justified since thrombin activity is known to be allosterically regulated [15, 19, 29]. This assumption additionally allows for the possibility that thrombin can be 'recycled', dissociating from a binding site upon fibrinopeptide cleavage and rebinding elsewhere to continue its enzymatic action.

**FpB cleavage.**   The stochastic binding model showed that bivalently bound thrombin remained in its bound state for long periods of time. In that model, thrombin was only taking part in binding reactions, not enzymatic ones. In the polymerization model, we assumed that upon enzymatic cleavage of FpB, bivalently bound thrombin would either release from the E-domain, staying bound to the $\gamma'$ site, or completely dissociate from fibrin and become free to rebind and cleave more fibrinopeptides in other locations. There is no experimental evidence to suggest that this dissocation upon cleavage occurs but this assumption alone led to model to give correct ordering of the protofibril number curves with respect to the $\gamma'$:E-domain ratio; without this assumption the curves were always in the wrong order because the $\gamma'$ binding sites in that scenario would act solely as a thrombin sink.

The parameter that controlled the possible fates of the bivalently bound thrombin after FpB cleavage, $R_{biv}$, was estimated to be 0.71 for the lowest error but the interquartile range of all the estimates encompassed fractions from 0.4 up to 0.8. An estimate of 0.71 suggests that 71% of bivalently bound thrombin was released directly into the free state upon FpB cleavage. This result has implications for another role of $\gamma'$ binding during fibrin polymerization. The bivalent binding plays two roles: it initially helps to localize thrombin to fibrin to enhance FpB

cleavage, but once all FpB is cleaved, the role switches to one of sequestration and protection from inhibition. Thus, early in clot formation fibrin binding is actively serving to localize thrombin to enhance polymerization via FpB cleavage, but later in the process, as FpB is depleted from various regions within a clot, fibrin is serving as a sink. Additionally, these results suggested that bivalent binding via $\gamma'$ likely contributes to thrombin trapping since protofibril aggregation is limited for fibrin I; once all fibrin in a protofibril has been converted to fibrin II, thrombin shifts to a high-affinity bound state and its time to removal increases enough to become physically trapped. Another way to say this could be that $\gamma'$ helps thrombin to stay actively converting fibrin I to II for as long as possible before it becomes sequestered and protected from inhibition.

**Parameter selection and FpB cleavage.** We estimated the rate that monomers associate, $k_{pi}$, to be about an order of magnitude larger than the rate that protofibrils grow in length, $k_{pg}$, suggesting that initiation of protofibrils is favored over growth of protofibrils. Interestingly, this is at odds with the previous modeling efforts of Weisel and Nagaswami, where $k_{pg} > k_{pi}$ to obtain long protofibrils and fibers. However, in that study, fibrinopeptide cleavage occurred in one step and occurred at a constant rate and thus it was the rate limiting step for initiation of protofibrils and thus protofibril growth had to overcome that rate. Here, we are explicitly modeling FpA and FpB cleavage using mass action kinetics, which led to a slower supply of $f_b$ and $f$, thus the association/initiation rate was higher as a counterbalance measure. Once the protofibrils were formed, the aggregation rates we estimated were similar to those from Weisel and Nagaswami, with estimates of the rate of protofibrils aggregating with other protofibrils, $k_{fi}$, to be about four orders of magnitude smaller than the rate at which protofibrils add to growing fibers, $k_{fg}$. This suggests that protofibrils are more likely to aggregate with growing fibers than with other protofibrils.

Factor XIII (FXIII), also known as fibrin stabilizing factor, is involved in covalent crosslinking of fibrin to stiffen the fibrin gel. In the absence of FXIII, fibrin dynamically remodels over time [44] and the presence of FpB might increase the likelihood of protofibril disassociation from fibers in that case. Our estimate of $S_{agg} = 0.0921$ means that fibrin I aggregated laterally at about 9% of the rate that fibrin II aggregated laterally. However, the model does not allow for FXIII crosslinking and so it is possible that it instead selects for a slower rate of lateral aggregation of protofibrils containing fibrin I. Explicitly accounting for FXIII cross-linking or the lack there of, may change this estimate considerably.

The parameter $S_{bind}$ incorporated the effects of steric inhibition of fibrin-fibrin binding due to the presence of a $\gamma'$ chain. The estimated value of 0.2524 means that the $\gamma'$ chain caused about a 75% reduction in the end-to-end binding rate. Allan et al. suggested that differences in turbidity seen between $\gamma_A/\gamma'$ and $\gamma_A/\gamma_A$ were due only to this steric inhibition, based on experiments with thrombin versus batroxobin [37]. Our model showed similar qualitative results (S1 Fig) where turning off FpB cleavage and decreasing the dissociation constant for E-domains to model batroxobin [45] led to a marked increase in protofibril number per fiber but also an increased lag time and overall decrease in protofibril number per fiber when comparing $\gamma_A/\gamma'$ to $\gamma_A/\gamma_A$. In this test case, the differences in protofibril number were entirely due to the steric inhibition. In turning off the steric inhibition ($S_{bind} = 1$), there was a collapse in the difference between the cases with batroxobin (no FpB cleavage) but not in those with thrombin (with FpB cleavage) since $S_{agg}$ still affected lateral aggregation with fibrin II in that case. Thus, our model results are in line with their experiments. However, our model also allowed us to investigate the effects of $\gamma'$ on lateral aggregation rates of fibrin I and fibrin II, which cannot be studied using batroxobin since fibrin II is not produced in its presence. Further experimental investigation would be useful to establish the relative contribution of steric inhibition and slowed lateral aggregation on fiber size with and without $\gamma'$ chains.

The scaling parameters for fibrinopeptide cleavage, $S_\alpha$, $S_\beta$, and $S_\gamma$, were estimated to be roughly the same and near one, which implies that parameter estimation selected rates in line with literature values. These estimates additionally imply that FpB cleavage is partially mediated by $\gamma'$ and that it occurs at roughly the same rate when thrombin is bivalently bound or bound only to an E-domain.

**Limitations and extensions.** Our model is complex, with dozens of species and interactions, and is still an oversimplification of the polymerization process. We used a continuum model to simulate a process that in reality has intricate structure and simply cannot be realized with this model. There are other mathematical approaches that aim to address questions about fiber structure that are based on coarse-grained molecular models [46, 47], but those studies did not investigate the interplay with $\gamma_A/\gamma'$ fibrin(ogen) and thrombin binding dynamics.

There are various ways that our model could be extended. For example, it would be interesting to see how different critical lengths of protofibrils or rates for oligomer-oligomer binding of various sizes would affect the fibrin fiber structure. Additional species of protofibrils could be added to allow for tracking branch points during fibrin formation, and we could also incorporate fibrinogen-fibrin-fibrinogen binding, which occurs when fibrinogen is in excess and can strongly inhibit the polymerization process [48].

As mentioned above, our model does not include the FXIII, which makes the polymerization process irreversible. FXIII can bind to either the $\gamma$- or $\gamma'$-chain of fibrinogen [49, 50] and can be converted to FXIIIa by thrombin while FXIII, thrombin, and fibrin are in a ternary complex. In this reaction, fibrin is thought to act as the cofactor [51], but the biochemical details of this are not fully understood. Since we do not consider FXIII, polymerization in our model is considered irreversible already but our model could be used as a platform to study such interactions. This would involve assumptions of reversibility, which would significantly increase the complexity, but is a potential future study to address questions related to $\gamma'$ and bivalent thrombin binding as well as thrombin trapping in the context of FXIII crosslinking.

In addition to reports that $\gamma'$ fibrinogen increases the risk of thrombosis, a recent study provided evidence that gamma' fibrinogen was associated with a lower risk of venous thromboembolism, ischemic stroke, cardioembolic stroke, and large artery stroke [52]. It is possible that $\gamma'$ fibrinogen may have different effects depending on the type of vascular disease. A full review of all the various roles for $\gamma'$ fibrinogen in the context of risk vs. protection of thrombosis is beyond the scope of this study, but some good reviews exist in the literature [17, 53]. These are very recent reports and the mechanisms underlying roles for risk vs. protection are not yet understood. Our suggested dual role of $\gamma'$ fibrinogen during polymerization may be able to help elucidate these mechanisms if combined with more spatial-temporal features. Future work could be done with this model to explore venous versus arterial flow conditions as well as injury severity in a spatial-temporal context [54, 55] to explore how these biochemical and biophysical features work together.

## Conclusion

Our mathematical modeling approach has suggested a dual role for $\gamma'$ during two distinct phases of the polymerization process: 1) $\gamma'$ effectively enhances thrombin activity during the early phase of polymerization; $\gamma'$ localizes thrombin to E-domains for FpA cleavage and dissociation upon cleavage allows for rapid rebinding and further cleavage of FpB, 2) $\gamma'$ sequesters thrombin during a later phase of polymerization; after all FpB has been cleaved the $\gamma'$-mediated bivalent binding acts as a sink for thrombin, enabling thrombin trapping within fibers and also providing protection from inhibition for long periods of time.

This is one of just a few mathematical models that exist of fibrin polymerization [56] and, to our knowledge, the only one that considers $\gamma'$ fibrinogen. Even though this mathematical model has its limitations, it has provided an experimentally testable idea and a new avenue of inquiry with respect to the dual role of thrombin-$\gamma'$ binding. Our mathematical modeling approach allowed for novel mechanistic insight into fibrin-thrombin interactions that may not have been apparent otherwise.

## Supporting information

**S1 Fig. Clot time ratios when simulating Batroxobin.** The ratios of the relative clot times between $\gamma_A/\gamma'$ to $\gamma_A/\gamma_A$ fibrinogen, with clot time being defined as the time to half-maximal turbidity. The model output (solid line) shows a similar trend but falls slightly below the experimental data from Kim et al. [38] (dots with error bars). To simulate the effects of Batroxobin, FpB cleavage was turned off and the dissociation constant for thrombin with the E-domain was decreased according to rates from the literature [45]. There was a marked increase in protofibril number per fiber, an increased lag time, and an overall decrease in protofibril number per fiber when comparing $\gamma_A/\gamma'$ to $\gamma_A/\gamma_A$.
(TIF)

**S1 Appendix. Mathematical equations describing the full fibrin polymerization model.**
(PDF)

## Author Contributions

**Conceptualization:** Michael A. Kelley, Karin Leiderman.

**Formal analysis:** Michael A. Kelley, Karin Leiderman.

**Funding acquisition:** Karin Leiderman.

**Investigation:** Michael A. Kelley.

**Methodology:** Michael A. Kelley, Karin Leiderman.

**Project administration:** Karin Leiderman.

**Software:** Michael A. Kelley.

**Validation:** Michael A. Kelley.

**Visualization:** Karin Leiderman.

**Writing – original draft:** Michael A. Kelley, Karin Leiderman.

**Writing – review & editing:** Michael A. Kelley, Karin Leiderman.

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
