## [Decision Letter · Decision Letter 0]

15 May 2022

Dear Dr. Leiderman,

Thank you very much for submitting your manuscript "Mathematical modeling to understand the role of bivalent thrombin-fibrin binding during polymerization" for consideration at PLOS Computational Biology. As with all papers reviewed by the journal, your manuscript was reviewed by members of the editorial board and by several independent reviewers. The reviewers appreciated the attention to an important topic. Based on the reviews, we are likely to accept this manuscript for publication, providing that you modify the manuscript according to the review recommendations.

The reviewers found your work to be well performed and a useful contribution to the understanding of fibrin-thrombin interactions during polymerization. Please consider their minor suggestions in the revised manuscript.

Sincerely,

Dimitrios Vavylonis

Guest Editor

PLOS Computational Biology

Mark Alber

Deputy Editor

PLOS Computational Biology

[LINK]

The reviewers found your work to be well performed and a useful contribution to the understanding of fibrin-thrombin interactions during polymerization. Please consider their minor suggestions in the revised manuscript.

Reviewer's Responses to Questions

**Comments to the Authors:**

Reviewer #1: The manuscript titled "Mathematical modeling to understand the role of bivalent thrombin-fibrin binding during

polymerization" presents a new method to simulate the fibrin polymerization that considers fibrin, fibrinogen and thrombin.

I have some minor concerns on the paper as below.

Comments:

1) The manuscript is nicely written and well explained.

2) A flowchart showing various methods and steps would have been of interest for the layman readers. This will help to understand the link between the SBM, polymerization and final results.

3) Despite of the tremendous effort that the authors put forward in this paper, the outcome of the model is not so appealing. Especially, when validating/comparing the results with the experiments, the comparison is made about protofibril number, distance between protofibrils through thrombin concentration and fibrinogen type. This means the spacio-temporal feature of polymerization process is absent, which is disappointing.

Reviewer #2: This is a rigorous, detailed computational study of fibrin-thrombin interactions during polymerization, with a focus on the role of gamma’ fibrinogen in altering the transient behavior of thrombin during clotting. The authors demonstrate that their model is able to capture and potentially explain the dual role of this fibrinogen variant, in which thrombin localization is first enhanced, leading to elevated polymerization, and then effectively quenched via sequestration in a latter phase. This is both interesting and useful contribution to the understanding of this process.

The work appears to have been carried out carefully and is described in detail in the manuscript. On this basis, I only have minor comments for the authors to consider in a revised version of the paper but otherwise recommend for publication in Plos Computational Biology with no reservations. Most of these are very minor in nature and the authors should be able to address them with little impact on the overall structure of the paper.

Specific Comments:

1. The risk vs. protection of the gamma’ chain is not fully resolved in the literature. The authors should cite and discuss some publications that found high gamma chain levels as protective in certain situations.

2. The authors have published very advanced models of thrombin generation with time. Were any simulations run where thrombin was initially not present but then generated with time?

3. Are the stochastic predictions in agreement with an earlier published estimate of apparent thrombin half-life in a fibrin clot exposed to flow to be about 1.1 hr (Chen, PLoS One, 2019).

4. The numerous equations both in the main text and the SI should be numbered so that they can be referenced and easily found by the readers.

5. Parsing through the modeling section with the various modeling components is rather challenging because it is not clear how the different model sub-components fit together. Can the authors either generate a separate summary figure for this, or use Fig. 2 to also annotate the model sections (numbers would help here as well)?

6. In the polymerization equations in the SI, can the authors verify that all factors related to possible double counting (i.e., i+j versus j+i) have been accounted for? Also, again here, numbering these equations and referring to them directly from the main text will help the interested reader more easily follow the details of the model.

**Have the authors made all data and (if applicable) computational code underlying the findings in their manuscript fully available?**

Reviewer #1: Yes

Reviewer #2: Yes

PLOS authors have the option to publish the peer review history of their article (what does this mean?). If published, this will include your full peer review and any attached files.

Reviewer #1: No

Reviewer #2: No

Figure Files:

Data Requirements:

Reproducibility:

References:

---

## [Decision Letter · Decision Letter 1]

19 Jul 2022

Dear Dr. Leiderman,

We are pleased to inform you that your manuscript 'Mathematical modeling to understand the role of bivalent thrombin-fibrin binding during polymerization' has been provisionally accepted for publication in PLOS Computational Biology.

Best regards,

Dimitrios Vavylonis

Guest Editor

PLOS Computational Biology

Mark Alber

Deputy Editor

PLOS Computational Biology

Reviewer's Responses to Questions

**Comments to the Authors:**

Reviewer #1: Thanks for addressing all the issues raised by the reviewers.

Reviewer #2: The authors have addressed the relatively minor concerns of both referees. The addition of the model flowchart (Fig. 2) certainly helps with navigating the discussion of the model. I recommend that the paper now be published in its present form.

**Have the authors made all data and (if applicable) computational code underlying the findings in their manuscript fully available?**

Reviewer #1: Yes

Reviewer #2: Yes

PLOS authors have the option to publish the peer review history of their article (what does this mean?). If published, this will include your full peer review and any attached files.

Reviewer #1: No

Reviewer #2: No

---

## [Editor Report · Acceptance letter]

29 Aug 2022

PCOMPBIOL-D-22-00553R1 

Mathematical modeling to understand the role of bivalent thrombin-fibrin binding during polymerization

Dear Dr Leiderman,

I am pleased to inform you that your manuscript has been formally accepted for publication in PLOS Computational Biology. Your manuscript is now with our production department and you will be notified of the publication date in due course.

With kind regards,

Agnes Pap
